# The patterns of soil nitrogen stocks and C:N stoichiometry under impervious surfaces in China

Qian Ding[1], Hua Shao[2], Chi Zhang[2, 1, 3, *], Xia Fang[4]

[1]Shandong Provincial Key Laboratory of Water and Soil Conservation and Environmental Protection, College of Resources and Environment, Linyi University, Linyi, 270600, China.
[2]State Key Laboratory of Desert and Oasis Ecology, Xinjiang Institute of Ecology and Geography, Chinese Academy of Sciences, Urumqi, 830011, China.
[3]Research Center for Ecology and Environment of Central Asia, Chinese Academy of Sciences, Urumqi, 830011, China.
[4]Xinjiang Institute of Engineering, Urumqi, 830091, China.

*Correspondence to*: Chi Zhang (zc@ms.xjb.ac.cn)

**Abstract.** Accurate assessment of soil nitrogen (N) storage and carbon (C):N stoichiometry under impervious surface areas (ISAs) is key to understanding the impact of urbanization on soil health and the N cycle. Based on 888 soil profiles from 148 sampling sites in 41 cities across China, we estimated the country's N stock (100 cm depth) in the ISA soil to be 98.74±59.13 Tg N with a mean N density ($N_{ISA}$) of 0.59±0.35 kg m$^{-2}$, which was significantly lower (at all depths) than the soil N density ($N_{PSA}$ = 0.83±0.46 kg m$^{-2}$) under the reference permeable surface areas (PSAs). The $N_{ISA}$ was also only about 53–69% of the reported national mean soil N density, indicating ISA expansion caused soil N loss. The C:N ratio of ISA (10.33±2.62) was 26–34% higher than that of natural ecosystems (forests, grasslands, etc.), but close to the C:N of PSA. Moreover, there was a significant C–N correlation in ISA soil, showing no signs of C–N decoupling as suggested by the previous studies. The ISA had smaller variances in the C:N ratio than did the PSA at regional scale, indicating convergence of soil C:N stoichiometry due to ISA conversion. The East subregion of China had the highest $N_{ISA}$, although its natural soil N density was the among lowest in the country. Unlike the vertical pattern in natural permeable soils, whose N density declined faster in the upper soil layers than in the lower layers, $N_{ISA}$ decreased linearly with depth. Similar to natural soil N, $N_{ISA}$ was negatively correlated with temperature; but unlike natural soil C:N which was positively correlated with temperature, the C:$N_{ISA}$ was negatively correlated with temperature. $N_{ISA}$ was not correlated with net primary productivity, but significantly correlated with the soil N density of adjacent PSA and the urbanization rate. These findings indicate the ISA soil had unique N distribution pattern, possibly as the result of intensive disturbances during land conversion.

## 1 Introduction

Nitrogen (N) is an essential nutrient that regulates ecosystem structure and function and maintains nutrient cycling (Fowler et al., 2013). It affects ecosystem carbon sequestration in various ways (Vitousek and Howarth, 1991), and the C:N stoichiometry conveys a rough measure of the mineralization and humification

of soil organic matter (SOM) (Chapin et al., 2011). Currently, global ecosystem structure and functions are intensively disrupted by urbanization, especially the rapid expansion of impervious surface area (ISA). The global ISA area in 2018 was 1.5 times larger than in 1990, at approximately $7.97 \times 10^5$ km$^2$ (Gong et al., 2020).

It has been suggested that the expansion of ISA blocks soil–atmosphere carbon/water exchanges, alters the physicochemical properties of soil, and seriously disrupts soil biogeochemical processes (Wei et al., 2014a; Yu et al., 2019). N loss from disturbed urban soils may cause groundwater N contamination (Li et al., 2022). Thus, there is an urgent need to study $N_{ISA}$ and SOC: $N_{ISA}$ patterns to provide a solid basis for assessing the potential risk of N loss and other negative impacts on urban ecosystems (Pereira et al., 2021).

Due to the difficulties in sampling beneath impervious surfaces, our knowledge about the biogeochemical processes in sealed soils is still very limited. Although the knowledge gap has gained more attention recently, most of the studies in ISAs have focused on the soil organic carbon (SOC) pool but have generally overlooked the soil N pool (Yan et al., 2015; Bae and Ryu, 2020; Cambou et al., 2018; Short et al., 1986). These studies showed that soil sealing not only causes a large amount of SOC loss but also alters the structure of the SOC

pools, indicating profound changes in soil carbon (C) processes (Wei et al., 2013; Raciti et al., 2012; Ding et al., 2022). To date, only a few isolated studies from seven cities (three in China, three in Europe and one in the USA) have reported the soil N content/density under ISA ($N_{ISA}$) (Table 1). All of these studies indicated that $N_{ISA}$ was lower than the N content/density ($N_{PSA}$) in pervious surface area (PSA). Two of them found extremely high C:N ratios in ISA (164 vs. 19 in PSA soil, 12.44 vs. 6.99 in PSA, respectively), suggesting

decoupling of C and N processes as the result of soil sealing (Raciti et al., 2012; O'riordan et al., 2021). A study from Nanjing city, China, however, found that ISAs had a lower C:N ratio than PSAs (Wei et al., 2014a).

Considering the high heterogeneity of urban soils, the available observations from 7 cities around the world are far from enough to provide useful information about the storage and characteristic distribution of $N_{ISA}$ at

large scale. In natural ecosystems, the distribution of N pools is significantly influenced by climate factors (Zhang et al., 2021). Temperature and precipitation are key drivers of soil biogeochemical processes (Wiesmeier et al., 2019). A previous study indicated that the ISA soil may also be affected indirectly by adjacent PSA (Yan et al., 2015), because many ISAs were converted from urban PSA during urban infilling (Delgado-Baquerizo et al., 2021; Kuang, 2019; Kuang et al., 2021). The soil organic matter input is

influenced by ecosystem net primary productivity (NPP) (Chan, 2001). The $N_{ISA}$ could also be correlated with the intensity of urbanization or human disturbances, which are influenced by population size, GDP, and the spatio-temporal patterns of built-up areas in a city (Bloom et al., 2008). Moreover, elevation and terrain may influence both the soil biogeochemical processes and ISA expansion (Zhu et al., 2022; Pan et al., 2023). Previous studies focused on individual cities, but regional scale surveys are required to investigate the

influences of climatic, ecological, geographic, and socioeconomic factors on $N_{ISA}$ distribution. Such information is not only necessary to evaluate global $N_{ISA}$ pool size, but also helpful in revealing the environmental-control mechanisms over the soil biogeochemical processes in ISA (Ding et al., 2022). For example, the urban ecosystem convergence theory suggests that cities from different regions tend to have

similar soil properties (e.g., SOC density) as a result of intensive human disturbances, even if their native soil

properties differ significantly (Pouyat et al., 2003). Regional soil surveys from multiple cities are required to evaluate this theory with soil nutrient data. In addition, more observational data are required to evaluate whether ISA soil has extremely high C:N ratio, which might indicate decoupling of soil C and N processes (Raciti et al., 2012; O'riordan et al., 2021).

Investigations on the vertical distribution pattern of soil N are also important, because the nutrient distribution

patterns through soil profiles are influenced by both natural and human factors. In natural ecosystems, vertical nutrient distributions are dominated by plant cycling relative to leaching, weathering dissolution, and atmospheric deposition, leading to nutrient concentrating in topsoil (Jobbágy and Jackson, 2001). Previous studies in urban areas, however, showed that the removal of plants and topsoil in the ISA may alter the vertical pattern of SOC, resulting in a more homogeneous SOC distribution through the soil profile (Yan et al., 2015;

Ding et al., 2022). Based on the observed SOC pattern, previous studies suggested that the changes in soil biogeochemistry in ISA was mainly caused by plant and topsoil removals and initial disturbance as opposed to postconstruction processes (Jobbágy and Jackson, 2001). Investigations on the vertical distribution patterns of $N_{ISA}$ can help us to evaluate this mechanism. However, most previous studies only sampled the topsoil or upper soil layers (Table 1) and thus could not obtain a complete picture of the vertical distribution pattern of

the $N_{ISA}$.

To address these issues, we investigated the patterns of China's $N_{ISA}$ pool and C:$N_{ISA}$ (C:N ratio of the ISA) based on 148 observations from 41 major cities across China (sampled at 100 cm depth and 20 cm intervals). The objectives of this study were to (a) compare $N_{ISA}$ with $N_{PSA}$, (b) reveal the spatial pattern of $N_{ISA}$ and C:$N_{ISA}$, and (c) identify the environmental factors correlated with $N_{ISA}$ and discuss the underlying mechanism.

We chose China as the study area because its urbanization rate is twice the global average, and approximately 2/3 of its urban area is occupied by ISA, which is also higher than the global average (Kuang, 2019). There are also relatively more previous $N_{ISA}$ studies in Chinese cities than in other countries (Table 1), which makes it easier to evaluate our results.

**Table 1: Compilation of soil $N_{ISA}$ studies**

| City, country | Previous studies | | | | This study | | | Background land-use type |
| --- | --- | --- | --- | --- | --- | --- | --- | --- |
| | Mean observed N density in the city (kg m$^{-2}$) | Mean observed N content in the city (g kg$^{-1}$) | Depth (cm) | References | Mean observed N density in the city (kg m$^{-2}$)$^*$ | Mean observed N content in the city (g kg$^{-1}$)$^*$ | Depth (cm) | |
| Beijing, China | NA | 0.61 | 0–10 | (Zhao et al., 2012) | 0.08±0.02 | 0.34±0.06 | 0–20 | Cropland and deciduous orchards |
| | NA | 0.54 | 10–20 | | | | | |
| | NA | 0.42 | 20–30 | | 0.09±0.02 | 0.4±0.11 | 20–40 | |
| | NA | 0.26 | 30–40 | | 0.09±0.02 | 0.4±0.11 | 20–40 | |
| | NA | 0.37 | 0–15 | (Hu et al., 2018) | 0.08±0.02 | 0.34±0.06 | 0–20 | |

| | | | | | | | | |
|---|---|---|---|---|---|---|---|---|
| Nanjing, China | NA | 0.49 | 0–20 | (Wei et al., 2014b) | 0.38±0.05 | 0.13±0.15 | 0–20 | |
| Yixing, China | 0.25 | NA | 0–20 | (Wei et al., 2014a) | 0.15±0.01 | NA | 0–20 | NA |
| New York, USA | 0.014 | NA | 0–15 | (Raciti et al., 2012) | 0.10±0.06 | NA | 0–15 | NA |
| Lancaster, UK | NA | 2.08 | 0–10 | (Pereira et al., 2021) | 0.07±0.04 | NA | 0–10 | NA |
| Greater Manchester, UK | 0.081 | NA | 0–10 | (O'riordan et al., 2021) | 0.07±0.04 | NA | 0–10 | NA |
| Toruń, Poland | 0.027 | 0.17 | 15–25 or 10–20 | (Piotrowska-Długosz and Charzyński, 2015) | 0.12±0.08 | NA | 0–20 | NA |

*±1SD

## 2 Materials and methods

### 2.1 Soil sampling

The soil samples were collected from 148 sample sites in 41 cities that were evenly distributed across mainland China except for the Tibetan Plateau during 2013–2014 (Figure 1). Depending on the city size,

multiple sample sites were identified in each city. Each site belonged to a separate city district, i.e., the soil samples were taken from 148 different city districts across China. The sample sites included various ISA types (roads, elevated piers, buildings, etc.) and PSA types (trees, shrubs, herbs, bare ground, vegetable plots, etc.). Detailed descriptions of the cities and sample sties can be found in (Ding et al., 2023).

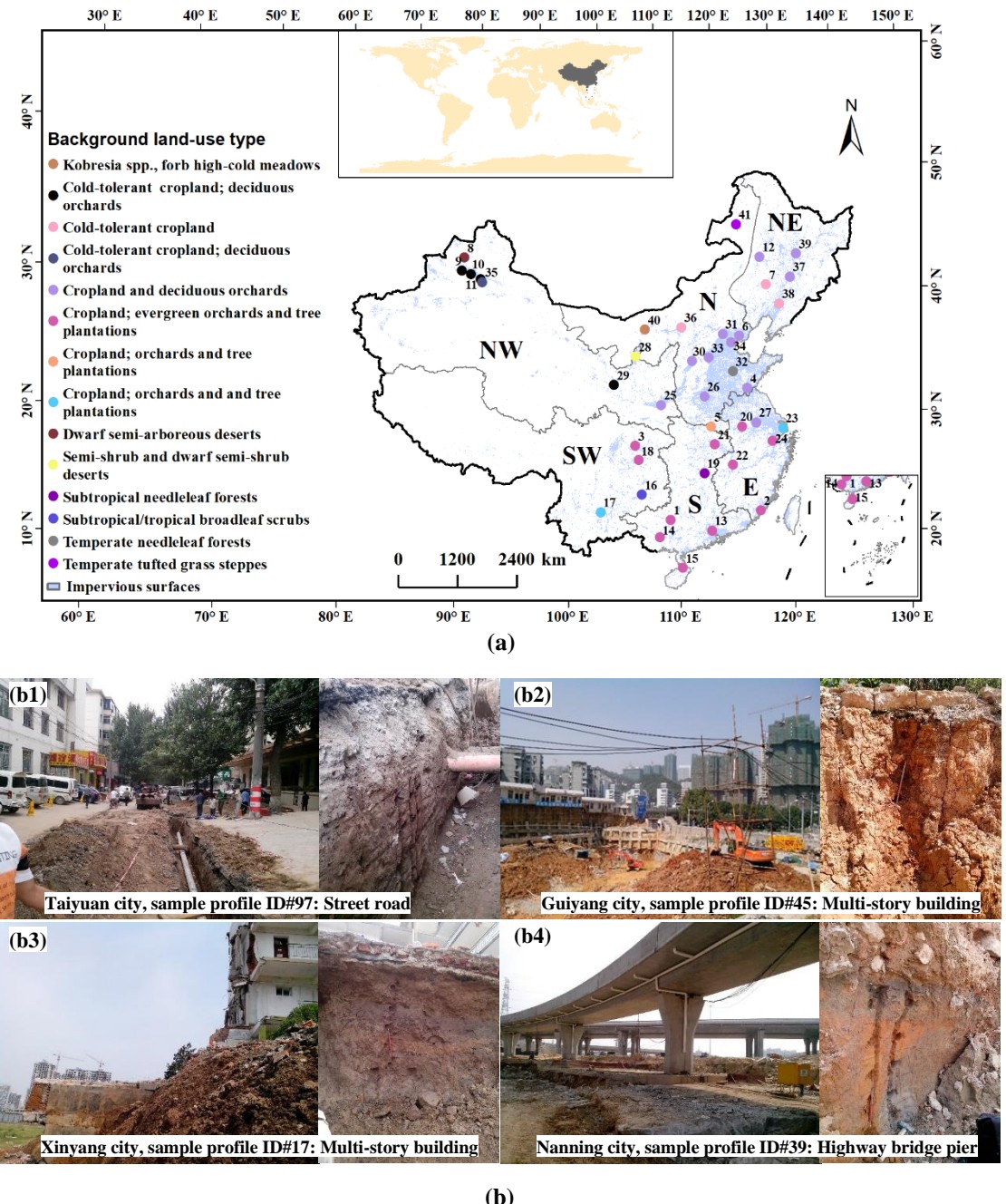

**Figure 1: Study area. (a) Spatial distribution of the sampled cities. The numbers in the map are the IDs of the studied cities, which can be used to retrieve detailed information of the sample sites from the online dataset of this study (Ding et al., 2023). The background land-use type shows the regional dominant land-use/land-cover type where the cities locate. To facilitate spatial analysis, we divided the country into six subregions – E: eastern China, S: southern China, N: northern China, NE: northeastern China, NW: northwestern China, SW: southwestern China. (b) Example photos for different sampling sites.**

At each sample site, 3 representative ISA sampling plots, more than 10 m apart from each other, were randomly selected. In addition, three paired sampling plots in adjacent PSAs were randomly selected for comparison. In each plot, a 100 cm depth profile pit was dug, and the soil profile was sampled at 20 cm intervals to the 100 cm depth with a 100 cm³ ring knife. Our study across China found that most of the Ekranic

(sealed) Technosol profiles have a clear boundary between the building material layer and the soil. Where the boundary is unclear, we treated the topsoil with a high amount of hard building materials, where artifacts >0.15 mm accounted for over half of the soil volume, as the building material layer. We only took samples in the soil below the building material layer. Samples with notable additions of anthropogenic artifacts, e.g., coal fly ash, mixed in the soil were discarded. Following the protocol of China's National Soil

Surveys, the visible non-soil artifacts in the remaining soil samples, such as fragmentations of bricks, glasses, stones, roots, etc., were picked out and discarded (Shi et al., 2004). A total of 4356 soil samples were eventually collected from 888 soil profiles. These samples (ID# XJBIZC0001–XJBIZC4356) are currently stored in the herbarium of the Xinjiang Institute of Ecology and Geography, Chinese Academy of Sciences.

To facilitate spatial analysis, we divided the country into six subregions – the northeast, north, northwest,

east, south, and southwest, according to geography, climate, and socioeconomics following Ding et al. (2022). To estimate the $N_{ISA}$ storage in each subregion, we multiple the mean $N_{ISA}$ density in the region with the region's ISA land area, which was derived from the 30 m resolution ISA map of mainland China (Zhang et al., 2020). Then, the $N_{ISA}$ stock of all subregions were added up to estimate the national $N_{ISA}$ storage.

## 2.2 Soil physical and chemical analyses

In this study, soil bulk density (BD) and N content were measured for each soil sample. Soil samples inside the ring knife were dried at 105 °C, and soil bulk weight (BD) (g cm⁻³) was measured, while the rest of the samples were air dried and passed through a 0.15 mm sieve, and the N content (g kg⁻¹) was measured by Kjeldahl digestion (Bremner and Mulvaney, 1982). The N density (kg m⁻²) per 20 cm soil layer was calculated according to Eq. 1, and the N density for the entire 100 cm profile was obtained by summing the

N density per 20 cm soil layer (Eq. 2).

$$N_i = \frac{NC_i \times BD_i \times 20}{100}, \tag{1}$$

$$N_{100cm} = \sum_{i=1}^{n} N_i, \tag{2}$$

where N represents N density (kg m⁻²), $i \in [1,5]$ represents soil layer (each 20 cm in thickness), NC is N content (g kg⁻¹), BD is soil bulk density (g cm⁻³).

## 2.3 Comparing $N_{ISA}$ and $N_{PSA}$, C:$N_{ISA}$ and C:$N_{PSA}$

A paired T test (2 tailed) was used to determine the difference between $N_{ISA}$ and $N_{PSA}$ and the difference between C:$N_{ISA}$ and C:$N_{PSA}$ (C:N ratio of the PSA). The C:N stoichiometry, i.e., the C:N ratio, shows the connection between the C process and N process. An extremely high C:$N_{ISA}$ in comparison with the reference C:$N_{PSA}$ indicates C–N decoupling due to soil sealing (Raciti et al., 2012).

The SOC density of the samples was reported in a previous study (Ding et al., 2022). We noticed that some research (Hu et al., 2018; Pereira et al., 2021; O'riordan et al., 2021) used the ratio between total C and total N to investigate the C:N stoichiometry in ISA soil. However, the content of soil inorganic C under impervious surfaces is likely altered by anthropogenic C from construction materials, and black C (the soot or carbonaceous products formed during the incomplete combustion of biomass and fossil fuels) (He and Zhang,

2009; Zhao et al., 2017; Zhu et al., 2019; O'riordan et al., 2021). In this study, we used the ratio between SOC and N to investigate the soil C:N stoichiometry, just like most soil studies in both ISA (Wei et al., 2014a; Raciti et al., 2012; Piotrowska-Długosz and Charzyński, 2015) and PSA (Lu et al., 2023; Schroeder et al., 2022; Yang et al., 2021).

We further investigate whether soil sealing may influence the variations of C:N stoichiometry at the national

scale. According to the urban ecosystem convergence theory, intensive human disturbances (e.g., soil sealing) could reduce variations in soil property at large scale (i.e., among different cities) even if the intensively disturbed areas may have similar or higher variations in soil properties at city scale compared to the less disturbed areas (e.g., PSA) (Pouyat et al., 2003). To evaluate this theory, we compared the mean inter-city C:N stoichiometry dissimilarity and the mean intra-city C:N stoichiometry dissimilarity between the ISA and

PSA. The inter-city dissimilarity (or regional scale dissimilarity) measured the Euclidean distance (Eq. 3) in C:N between each pair of different cities, while the intra-city dissimilarity (or local scale dissimilarity) measured the Euclidean distance in C:N between each pair of sampling sites within the same city, all combinations included. If the urban ecosystem convergence theory was correct, we expect to see ISA having lower inter-city C:N dissimilarity than PSA, but higher or similar intra-city C:N dissimilarity than/to PSA.

$$\text{Euclidean distance} = \sqrt{\left(C{:}N_i - C{:}N_j\right)^2},\tag{3}$$

where $C{:}N_i$ and $C{:}N_j$ are the soil C:N ratios of site $i$ and site $j$, respectively, when measuring the intra-city dissimilarity, or the city-averaged C:N ratios of city $i$ and city $j$, respectively, when measuring the inter-city dissimilarity.

### 2.4 Investigating the vertical pattern of $N_{ISA}$

Unlike other studies that focused on topsoil, our multiple–layer soil sampling data made it possible to study the vertical pattern of $N_{ISA}$ to a 100 cm depth. The proportions of N stored in the 0–20 cm depth, 0–40 cm depth, 0–60 cm depth, and 0–80 cm depth to the total (100 cm depth) N stock in each sample profile were calculated and plotted against the soil depth to reveal the vertical distribution pattern of $N_{ISA}$ and $N_{PSA}$. Based on these data, we could model how N storage changed with soil depth. According to Yang et al. (2007), 46%

of the N stock (in 1 m depth) of natural soil is stored in the top 0–30 cm soil, and 68% of the N stock is stored in the top 0–50 cm, translating into a power function fitting model:

$$\text{N}_{\text{Natural}}\%_d = -0.0074d^2 + 1.7378d = (1.7378 - 0.0074d) \times d,\tag{4}$$

where $N_{Natural}\%_d$ is the proportion of total N stock (in 100 cm depth) stored to depth $d$ cm in natural soil in China. The equation shows that the $N_{Natural}\%_d$ does not increase linearly with soil depth, its increasing rate

(i.e., $1.7378 - 0.0074d$) reduces with soil depth $d$. This pattern indicates the natural soil N does not have homogeneous N density through the soil profile, it decreases with depth.

**2.5 Correlation analysis between $N_{ISA}$ and potential environmental factors**

Our large scale soil survey made it possible, for the first time, to investigate the correlations between soil N and various environmental factors so as to identify the climatic, ecological, geographical and socio-economic

factors that may control or influence the N and C:N dynamics in sealed soil. We selected 15 indicators to investigate the factors associated with $N_{ISA}$, including mean temperature, annual precipitation, background NPP (averaged natural ecosystem NPP in a 5 km buffer outside the city), $C:N_{PSA}$ and $N_{PSA}$, longitude, latitude, elevation, population density, built-up area in a city, urbanization rate as indicated by the fraction of the built-up area that expanded after 2002, ISA coverage in built-up areas, greenspace coverage in built–up areas, per

capita greenspace, city GDP, and per capita GDP. We also investigated the correlation between soil BD and the $N_{ISA}$ content.

Gridded datasets of environmental factors, including mean annual temperature (Figure 2a), annual precipitation (Figure 2b), and elevation (Figure 2d) at 1 km resolution, were obtained from the Data Center for Resource and Environmental Sciences, Chinese Academy of Sciences (http://www.resdc.cn/). The

national NPP (1985–2015) estimates at 1 km resolution was obtained from the Digital Journal of Global Change Data Repository (https://www.geodoi.ac.cn/) (Figure 2c). Statistical datasets include the Ministry of Housing and Urban–Rural Development of China (www.mohurd.gov.cn/) urban built-up area, population density, built-up area green space rate, per capita built-up area green space, the National Bureau of Statistics of China (data.stats.gov.cn/) total urban GDP, and per capita GDP. We used correlation analysis to investigate

the relationships. If the variables were normal distributed and linear correlated, then the Pearson's correlation (2 tailed) were applied. Otherwise, Spearman's correlation (2 tailed) were used.

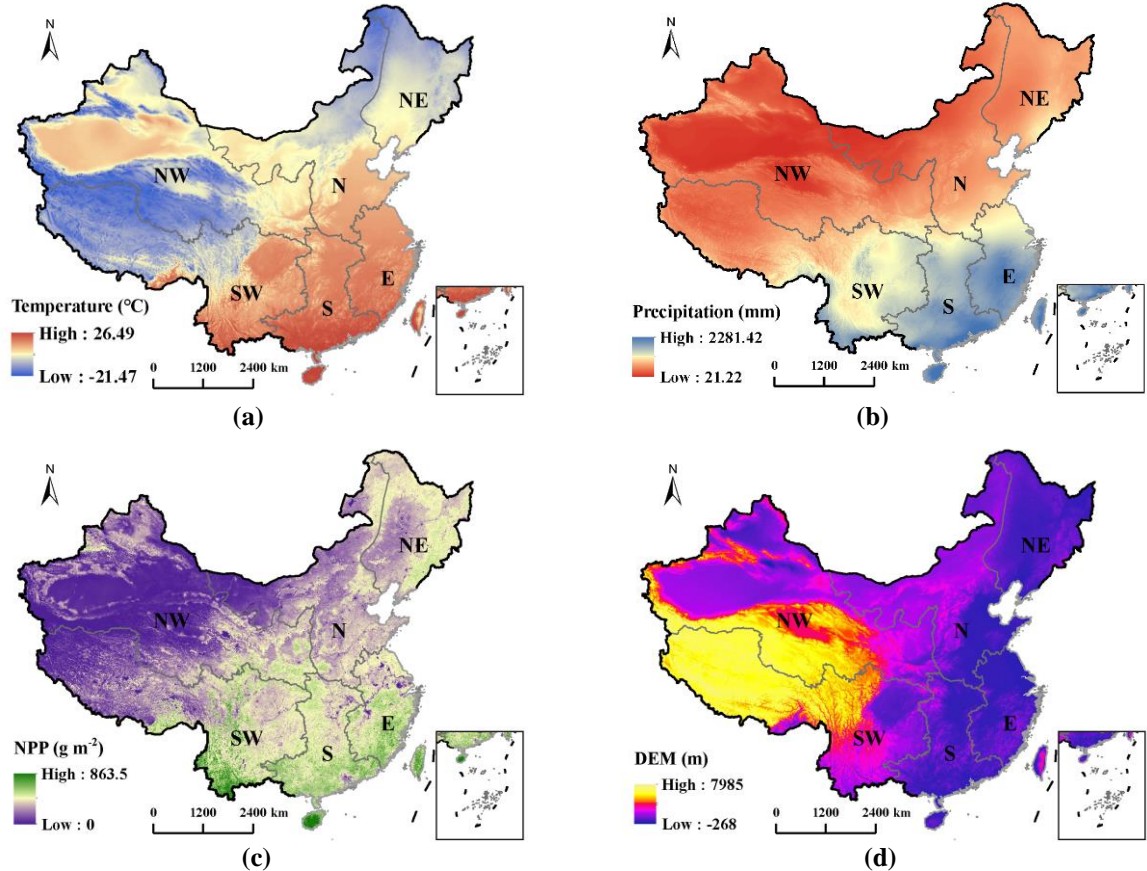

**Figure 2: A subset of the spatial datasets of climatic, ecological and terrain factors whose correlations with $N_{ISA}$ were investigated in this study, including (a) annual precipitation normal (1981–2010), (b) air temperature normal (1981–2010), (c) mean annual NPP (1985–2015), and (d) digital elevation model (DEM). E: eastern China, S: southern China, N: northern China, NE: northeastern China, NW: northwestern China, SW: southwestern China.**

## 3 Results

### 3.1 N densities and storage under ISA in China

The national mean $N_{ISA}$ density in the 100 cm soil profile was 0.59±0.35 kg m$^{-2}$ (mean±1 SD), ranging from 0.08–1.88 kg m$^{-2}$ with a median value of 0.48 kg m$^{-2}$. Paired t tests showed that the $N_{ISA}$ was significantly (approximately 30%) lower (P<0.01) than the reference $N_{PSA}$ (Figure 3a). Moreover, the $N_{ISA}$ was lower than the reference $N_{PSA}$ at all soil depths and in all subregions of China (Figure 4a). The national total $N_{ISA}$ stock was about 98.74±59.13 Tg N.

C:$N_{ISA}$ (10.33±2.62) was significantly lower than C:$N_{PSA}$ (10.93±3.19) (Figure 3b). Moreover, $N_{PSA}$ and SOC$_{PSA}$ were significantly correlated (R=0.893, P<0.01), and $N_{ISA}$ and SOC$_{ISA}$ were also significantly correlated (R=0.926, P<0.01). There were no signs of C–N decoupling according to our data.

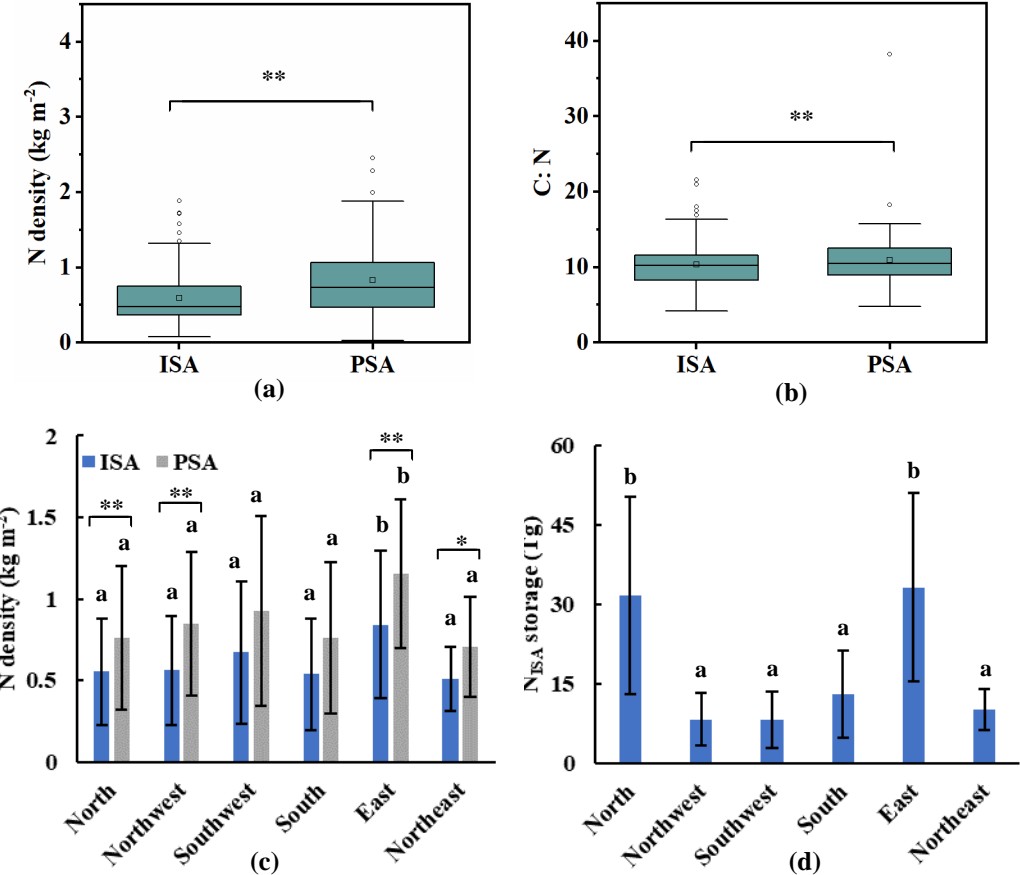

**Figure 3: The N density, N storage and C:N ratio of the ISA in China. The soil N density and C:N in the ISA and the reference PSA are compared in (a) and (b), respectively. The square box shows median and quad values, the inner small rectangle is the mean value. The regional mean $N_{ISA}$ density and $N_{PSA}$ density in different subregions are compared in (c). The regional $N_{ISA}$ storage of different subregions are compared in (d). The letters indicate the significance of the difference among the subregions. * and ** indicate significant differences between ISA and PSA, p<0.05 and p<0.01, respectively.**

### 3.2 Spatial variation and spatial trend analysis

To facilitate spatial analysis, we divided the country into six subregions – the Northeast, North, Northwest, East, South, and Southwest, according to geography, climate, and socioeconomics (Ding et al., 2022) (Figure 1a). The highest $N_{ISA}$ density of 0.84±0.45 kg m$^{-2}$ was found in the East, while the lowest $N_{ISA}$ density of 0.51±0.20 kg m$^{-2}$ was found in the South (Figure 3c). Notably, the urban soil N densities, both in ISA and PSA, in the East were significantly higher than that of the other subregions except for the Southwest where the variations of soil N densities were extremely high (Figure 3c). As the result, the East subregion accounted for the largest share (34%) of the $N_{ISA}$ stock in China (Figure 3d).

Figure 4 shows the ISA soil samples had lower inter-city C:N dissimilarity (1.86±1.40 vs. 2.43±2.15) than PSA, but similar intra-city C:N dissimilarity to PSA. This pattern indicates that although the ISA soil and the PSA soil had similar variations in C:N stoichiometry at the local scale (within a city), the C:N variations at

**

national scale (among the cities) were reduced for the ISA soil, possibly due to the intensive human

disturbances on ISA soil as predicted by the urban ecosystem convergence theory (Pouyat et al., 2003).

The spatial trend analysis of the $N_{ISA}$ showed a slow decline followed by a rapid increase in the north–south direction and a rapid decline followed by a rapid increase in the east–west direction (Figure 5a); the spatial trend analyses of the $N_{PSA}$ produced similar concave lines in both the north–south and the east–west directions but with a more drastic initial decline in the north–south direction and a flatter trend in the east–west direction

(Figure 5b). The spatial trend analysis of $C:N_{ISA}$ showed a rapid increase in the north–south direction and a rapid decrease followed by a slow increase in the east–west direction (Figure 5c); the spatial trend analysis of $C:N_{PSA}$ showed a slow increase in the north–south direction and a slow decrease in the east–west direction (Figure 5d). According to the spatial trend analyses, the change rate of $C:N_{ISA}$ was significantly higher than that of $C:N_{PSA}$, which was consistent with the results of inter–city C:N ratio dissimilarity analysis (Figure 4).

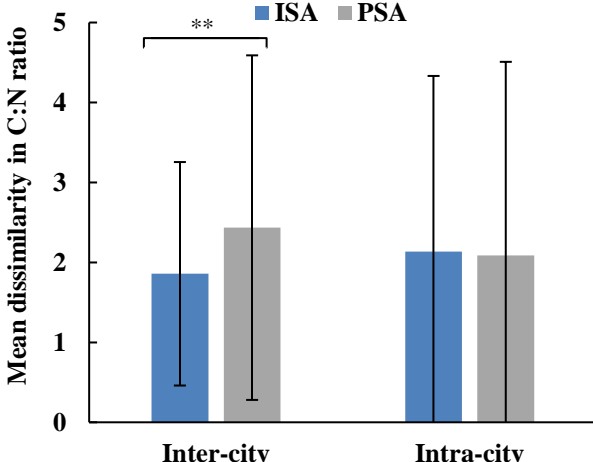

**Figure 4: Comparing the inter–city variation and the intra–city variation of C:N ratios between the ISA and PSA. The variations were measured by the dissimilarity of (or the Euclidean distance between) paired observations. For intra–city variation, the soil C:N dissimilarity between each pair of different sampling sites within the same city were calculated and averaged; for inter-city variation, the soil C:N dissimilarity between each pair of different cities under investigation were calculated and averaged. \*\***

**indicate significant differences between ISA and PSA (p<0.01).**

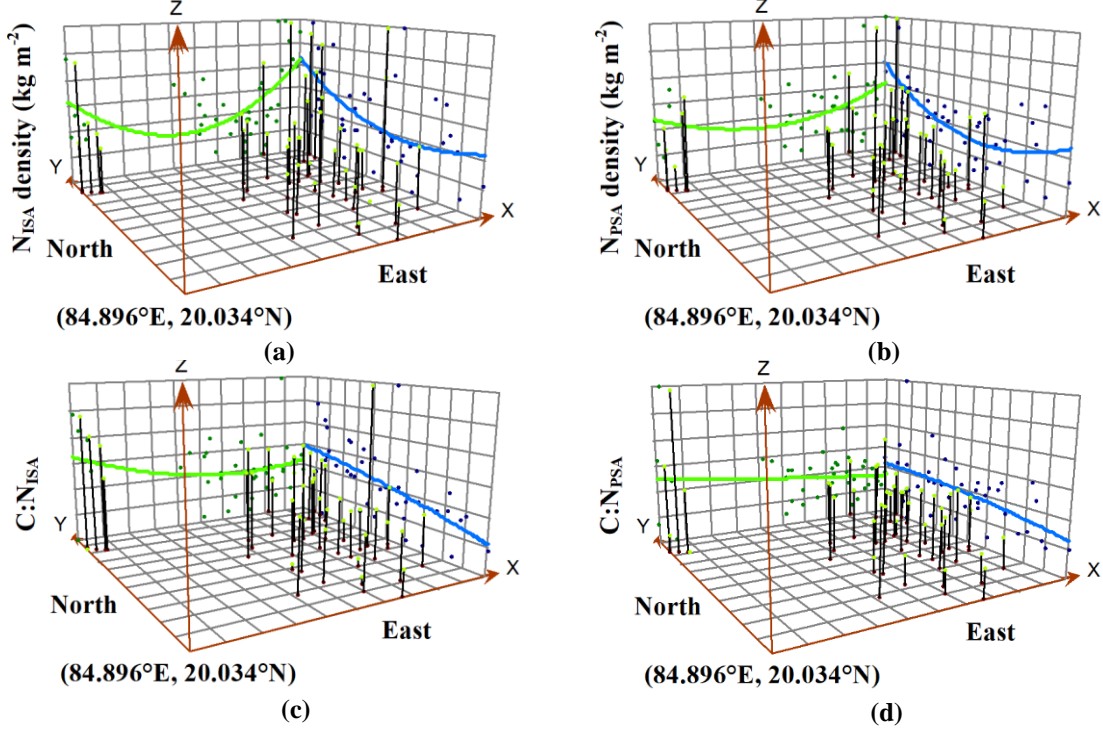

**Figure 5: Trend analysis on the variations of the city-level mean (a) $N_{ISA}$ density, (b) $N_{PSA}$ density, (c) C:$N_{ISA}$ and (d) C:$N_{PSA}$, in the east–west direction (the green trend line) and the north-south direction (the blue trend line) across China. The locations of sampled cities are plotted on the x, y plane. The x-axis indicates the east-west direction; the y-axis indicates the north-south direction. Above each sampled city, its city-averaged value of observed soil N property (i.e., $N_{ISA}$ density or $N_{PSA}$ density or C:N ratio) is given by the height of a stick in the z-dimension. The values are projected onto the x, z plane (i.e., the left vertical plane) and the y, z plane (i.e., the right vertical plane) as scatterplots. This can be thought of as sideways views through the three-dimensional data. Second-order polynomials are fit through the scatterplots on the projected planes. The green line in the x, z plane shows the trend of value variation in the east-west direction, while the blue line in the y, z plane shows the trend of value variation in the north-south direction.**

### 3.4 Vertical distribution pattern of $N_{ISA}$

In this study, the vertical profiles of soils under ISAs were systematically sampled and analysed at 20 cm intervals to a 100 cm depth, and the storage of $N_{ISA}$ increased linearly with soil depth (Figure 6). This linear distribution pattern was evident at the national scale ($R^2 = 1$, $P < 0.001$), and the vertical distribution pattern of $N_{ISA}$ in China can be described by a linear model (Eq. 5):

$$N_{ISA}\%_d = 1.0324d, \tag{5}$$

where $N_{ISA}\%_d$ is the percentage of the N stock (to 100 cm depth in total) located in the top $d$ (cm) depth of the soil.

**Proportion of the total N storage (%)**

$N_{Natural}\% = -0.0074d^2 + 1.7378d$
$R^2 = 1, p<0.01$
(data source: Yang et al., 2007)

$N_{ISA}\% = 1.0324d$
$R^2 = 1, p<0.01$
(data source: this study)

◇ $N_{ISA}$
△ $N_{Natural}$

**Figure 6: Comparing the vertical distribution pattern of N between the sealed soil ($N_{ISA}$) and the natural soil ($N_{Natural}$) in China (refer to Section 2.4 Equation 4).**

### 3.5 The natural and socioeconomic factors correlated with $N_{ISA}$ and C:$N_{ISA}$

Latitude, temperature, NPP, and C:$N_{ISA}$ were non-normally distributed, so we analysed them using Spearman's correlation (2 tailed), and the correlations of the remaining variables were analysed using Pearman's correlation analysis (2 tailed). The impacts of climate and geographic factors were confirmed by correlation analyses, which showed $N_{ISA}$ to be negatively correlated with temperature (R=–0.486) (Table 2). In addition, the $N_{ISA}$ had a positive correlation with the N of urban PSA (R=0.715) and a negative correlation (R=–0.34) with the urbanization rate as indicated by the fraction of the newly expanded ISA since 2002 (i.e., $f_{new\_ISA} = \frac{ISA_{2015} - ISA_{2002}}{ISA_{2015}}$). Surprisingly, we did not find significant correlations between $N_{ISA}$ and common environmental drivers like precipitation and NPP at 95% significant level, although the $N_{ISA}$ had a weak negative correlation with precipitation (R=–0.268) at 90% significant level.

The C:$N_{ISA}$ was negatively correlated with both precipitation (R=–0.620) and temperature (R=–0.561), but positively correlated with latitude (R=0.513) (Table 2). In addition, the C:$N_{ISA}$ had a positive correlation with the N of urban PSA (R=0.515) and a negative correlation (R=–0.516) with the NPP.

**Table 2: Correlations between $N_{ISA}$, C:$N_{ISA}$ and potential environmental drivers**

| Factors | N density (kg m$^{-2}$) | | C:$N_{ISA}$ | |
|---|---|---|---|---|
| | Correlation Coefficient | Sig. (2 tailed) | Correlation Coefficient | Sig. (2 tailed) |
| Longitude | 0.196 | 0.22 | –0.186 | 0.24 |
| Latitude | 0.275 | 0.08 | 0.513[**] | 0.00 |
| DEM (m) | 0.141 | 0.38 | 0.477[**] | 0.00 |

| | | | | |
|---|---|---|---|---|
| Annual precipitation (mm) | –0.268 | 0.09 | –0.620** | 0.00 |
| Mean Temperature (℃) | –0.486** | 0.00 | –0.561** | 0.00 |
| NPP (g m$^{-2}$) | –0.106 | 0.51 | –0.516** | 0.00 |
| ISA coverage in built-up area (%) | –0.126 | 0.43 | –0.171 | 0.29 |
| Built-up area (km$^2$) | –0.072 | 0.65 | 0.062 | 0.70 |
| Greenspace coverage in built-up area (%) | –0.229 | 0.15 | –0.063 | 0.69 |
| Population density (person km$^{-2}$) | –0.032 | 0.84 | –0.072 | 0.66 |
| Per capita GDP (person $10^{-4}$ yuan) | –0.012 | 0.94 | –0.145 | 0.37 |
| City GDP (billion yuan) | –0.015 | 0.93 | –0.200 | 0.21 |
| Per capita greenspace (m$^2$) | 0.098 | 0.54 | 0.044 | 0.79 |
| The fraction of the newly expanded ISA since 2002 (%) | –0.340* | 0.03 | –0.197 | 0.22 |
| $N_{PSA}$ density (kg m$^{-2}$) | 0.715** | 0.00 | NA | NA |
| C:$N_{PSA}$ | NA | NA | 0.515** | 0.00 |
| BD | –0.104 | 0.52 | NA | NA |

*$p < 0.05$;
**$p < 0.01$.

## 4 Data availability

The dataset "Observations of soil nitrogen and soil organic carbon to soil nitrogen stoichiometry under the
impervious surfaces areas (ISA) of China" includes N density, N content, BD, C:N, and other related data under ISA and PSA. It also contains geographical coordinates of sampling locations, as well as spatial pattern layer files of $N_{ISA}$ density and C:$N_{ISA}$ in China. This dataset is available from the National Cryosphere Desert Data Center (www.ncdc.ac.cn/portal/metadata/review/04cee3f5-64bb-4b22-9368-ee1c55f9c2bb?lang=en) (Ding et al., 2023).

## 5 Discussion

### 5.1 Comparing the N density and C:N stoichiometry in ISA soil with those in natural soils

Our results were comparable to or moderately lower than the previously reported topsoil (0–20 cm) $N_{ISA}$ contents/densities in Chinese cities, including Beijing (0.34±0.06 g kg$^{-1}$ vs. 0.37–0.61 g kg$^{-1}$) (Hu et al., 2018; Zhao et al., 2012), Nanjing city (0.13±0.15 g kg$^{-1}$ vs. 0.49 g kg$^{-1}$) (Wei et al., 2014b) and Yixing city (0.15±0.01 kg m$^{-2}$ vs. 0.25 kg m$^{-2}$) (Wei et al., 2014a) (Table 1). Our observed $N_{ISA}$ content (0.4 g kg$^{-1}$) in the 20–40 cm soil layer in Beijing was also comparable to the reports by Zhao et al. (2012) (0.26–0.42 g kg$^{-1}$). Outside China, the reported topsoil (0–10 cm) $N_{ISA}$ density in Greater Manchester, UK (0.081 kg m$^{-2}$), was comparable to our estimated mean $N_{ISA}$ density in 0–10 cm (0.07±0.04 kg m$^{-2}$) in China (O'riordan et

al., 2021), but the reported $N_{ISA}$ in New York, USA and Toruń, Poland, was much lower than our results and the reports from other Chinese city studies (Table 1) (Raciti et al., 2012; Piotrowska-Długosz and Charzyński, 2015).

Compared with previous assessments of China's soil N stock, $N_{ISA}$ accounted for 0.96–1.47% of the total soil N pool in China (Zhang et al., 2021; Xu et al., 2020). The $N_{ISA}$ pool size (98.74±59.13 Tg) exceeded the vegetation N of scrubland (8.1–50 Tg) (Xu et al., 2020) and grassland (48.8 Tg) (Zhang et al., 2021) in China. The $N_{ISA}$ density (0.58±0.12 kg m$^{-2}$) was lower than that of natural soil and equivalent to 53–69% of the national average (Yang et al., 2007; Xu et al., 2020). The N densities in ISA soil (0.59±0.35 kg m$^{-2}$) were lower than those in other ecosystems, such as forest (1.29 kg m$^{-2}$), farmland (1.13 kg m$^{-2}$), and grassland (1.11 kg m$^{-2}$) (Xu et al., 2020), indicating that ISA construction resulted in N loss. Previous studies ignored the impacts of ISAs (Tian et al., 2006; Yang et al., 2007) and thus might have overestimated China's soil N pool size.

Our estimated C:$N_{ISA}$ (10.33±2.62) in China matched the previously observed C:$N_{ISA}$ (10.8) in Yixing city, China (Wei et al., 2014a). It has been suggested that different terrestrial ecosystems may have similar C:N ratios (Yang et al., 2021). This study showed that the C:$N_{ISA}$ (10.33±2.62) was only slightly lower than the C:$N_{PSA}$ (10.93±3.19) in urban ecosystems but much higher than the C:N ratios of natural ecosystem soils such as forests (8.21), croplands (8.18), and grasslands (7.7) (Xu et al., 2020; Tang et al., 2018). Therefore, it is possible that the C:N stoichiometry could remain relatively stable within the same land–use type (natural ecosystems, urban areas, etc.) but might differ significantly among different natural or human–disturbed land–use types.

### 5.2. The C:N stoichiometry analysis showing no sign of C–N decoupling in the ISA soil

The above comparison indicates that ISA soil has a higher C:N than natural soils. A study in New York City reported that the N density in the ISA was 95% lower than that in the PSA, leading to an extremely high soil total C:total N ratio (Raciti et al., 2012; Majidzadeh et al., 2017). Therefore, Raciti et al. (2012) suggested that paving decouples the C and N cycles. Our observations, however, showed that the soil N of ISA was only 30% lower than that of PSA, and the C:$N_{ISA}$ was lower than the C:$N_{PSA}$ in China. Furthermore, there was a significant positive correlation (R=0.926, P<0.01) between N and SOC in ISA soil. Similarly, Wei et al. (2014a) found that C:$N_{ISA}$ was lower than C:$N_{PSA}$ in Yixing city, China, and O'Riordan et al. (2021) found a significant positive correlation between N and C in ISA soil in Greater Manchester, UK, even though they also observed an increased total C:total N ratio in ISA soil compared to PSA soil. There were no signs of C–N decoupling according to our data and others (O'riordan et al., 2021; Wei et al., 2014a). It is possible that the extremely high C:N ratio observed in previous studies might be merely caused by anthropogenic C inputs that partially compensated for SOC loss during land conversion (O'riordan et al., 2021). Because the construction materials could add large amounts of inorganic C into soil, it is preferable to investigate the C:N stoichiometry under ISAs with the C:N ratio rather than the total C:total N ratio. This study highlights the

important role of N in urban biogeochemical research, which helps to prevent us from being confused/misled
by the complex C dynamics in urban soil due to anthropogenic C inputs.

**5.3 Potential driving factors of the $N_{ISA}$ and C:$N_{ISA}$**

The spatial distribution pattern of soil N was significantly correlated with climate factors such as temperature and precipitation in natural ecosystems (Yang et al., 2007). In general, the soil N in China's temperate and subtropical ecosystems were negatively correlated with temperature (Lu et al., 2017). Similarly, our study found a negative correlation between $N_{ISA}$ and temperature. There was no significant correlation between precipitation and the soil N in natural ecosystem, except for dryland where a positive correlation has been found (Lu et al., 2017). We didn't find significant corelation between precipitation and $N_{ISA}$ at the 95% confidence level, although there was a weak negative correlation at the 90% confidence level. Previous study showed the $SOC_{ISA}$ was also negatively correlated with precipitation, and it was suggested that the observed soil biogeochemistry (SOC, nutrient content etc.) under impervious surface was mainly determined by the losses (esp. in topsoil) during land conversion (Majidzadeh et al., 2018; Cambou et al., 2018; Edmondson et al., 2012). Higher precipitation leads to higher soil nutrient loss during land conversion (Ding et al., 2022). The relatively weak correlation between $N_{ISA}$ and precipitation (compared with the correlation between $SOC_{ISA}$ and precipitation) as well as the negative correlation between C:$N_{ISA}$ and precipitation might indicate that the N loss during land conversion was not as significant as the loss in SOC. It is also possible that the high N deposition in urban ISA might somehow replenish the $N_{ISA}$ pool.

$N_{ISA}$ was not correlated with background NPP but positively correlated with the soil N in the adjacent urban PSA. This pattern agrees with the previous report that the $SOC_{ISA}$ was mainly influenced by the SOC in the adjacent urban PSA rather than the background SOC and NPP (Ding et al., 2022). However, there was a negative correlation between C:$N_{ISA}$ and background NPP.

The soil C:N ratio could be a more stable parameter (Yang et al., 2021). Tian et al. (2010) found the soil C;N ratio was relatively stable among climate zones in rural ecosystems in China. It has been observed that the soil stoichiometric characteristics in China are influenced by geographical parameters such as altitude and latitude (Sheng et al., 2022). Lu et al. (2023) found a lower C:N ratio at higher latitudes in China, suggesting a positive correlation between C:N and temperature in natural ecosystem soils. Our study, however, found that the C:$N_{ISA}$ ratio increased with latitude and that there was a significant negative correlation between the C:$N_{ISA}$ ratio and temperature. The soil C:N ratio of natural ecosystems is influenced by plant litter input and N uptake. Ecosystems in warmer regions have higher NPP, resulting in higher inputs of litter with a high C:N ratio (compared with the soil C:N ratio) and higher N uptake by roots, thus reducing soil inorganic N. Therefore, the C:N ratio is positively correlated with temperature in natural ecosystems. However, the C:N ratio under the impervious surface is solely determined by the relative mineralization rate of C and N. It seems that soil ecosystems have a higher retention capacity for N than for C (C fixation is unlikely to be found in sealed soil). Therefore, while both the soil $N_{ISA}$ pool and the $SOC_{ISA}$ pool decrease when the

temperature increases, the net N mineralization rate is lower than the C mineralization rate, leading to a
negative correlation between the C:$N_{ISA}$ ratio and temperature.

**5.4 $N_{ISA}$ had different regional distribution pattern and vertical distribution pattern from rural soil N**

Our study and Tian et al. (2006)'s study on China's soil N had same subregion zone design. However, we found the urban soil (both the ISA and PSA) in the East zone had the highest N density while Tian et al. (2006) found the rural soil N density in the East zone was among the lowest in the country. The relatively
high precipitation and temperature in the East China may lead to high SOM decomposition rate and nutrient leaching rate, which explains its low rural soil N density (Tian et al., 2010). However, the East region was also the most developed region in China for the last several centuries. Its cities had high population density and long urbanization history. The long-term intensive human activities might leave profoundly footprint in the soil biogeochemical processes, significantly elevated its N content. This finding, together with the
relatively low inter-city C:N variations/dissimilarities in the ISA (see section 3.2), indicate intensive human disturbances might override the nature environmental effects in shaping regional distribution pattern of soil N processes, further confirmed the urban ecosystem convergence theory (Pouyat et al., 2003).

The vertical N distribution patterns are also different between the ISA soil and rural soil. Unlike the vertical distribution pattern of natural soil N, which decreased with depth and should be modelled with a second-
degree polynomial fitting function (Eq 4), the vertical distribution of $N_{ISA}$ was relatively homogeneous through the soil profile and could be modelled with a linear function (Eq 5; Figure 6). The SOC under the impervious surface had similar vertical distribution pattern (Ding et al., 2022). The unique vertical pattern reflects the effect of human disturbance during land conversion, which led to topsoil soil organic matter and nutrient loss and reduced the SOC and N gradient through the soil profile (Majidzadeh et al., 2018; Cambou
et al., 2018; Edmondson et al., 2012).

**5.5 Potential applications of the data**

N plays an important and complex role in natural ecosystems. Soil microbial C use efficiency is negatively correlated with C:N (Schroeder et al., 2022). Incorporating N into Earth system models can improve the accuracy of C cycle estimates (Fleischer et al., 2019), and a good description of N can help understand and
predict the patterns and mechanisms of global C dynamics (Zhang et al., 2021) and provide a reliable basis for exploring how geochemical cycles are coupled.

For a long time, knowledge of biogeochemical cycles under impervious surfaces has been a major gap in urban biogeochemical research. Until recently, the size and pattern of $N_{ISA}$ pools and their contributions to the global N cycle have remained unclear. Our research, which is the first national–scale study on $N_{ISA}$ and
C:$N_{ISA}$, helps to fill this gap by improving our understanding of the special pattern of soil N under impervious surfaces. Such information is necessary when assessing urbanization impacts on global C and N cycles (Lorenz and Lal, 2009).

**Author contributions.** Conceptualization: CZ. Data curation: CZ. Formal analysis: QD. Funding acquisition: ZC and HS. Investigation: HS. and XF. Methodology: QD and CZ. Project administration: CZ and HS. Resources: HS. and XF. Software: QD. Supervision: CZ. Validation: QD. Visualization: QD. Writing – original draft preparation: QD. Writing – review & editing: QD., HS. and CZ.

**Competing interests.** The authors declare that they have no conflicts of interest.

**Acknowledgements.** We want to thank the reviewers for their constructive comments which are helpful for improving our article. This project was funded by the Strategic Priority Research Program of the Chinese Academy of Sciences (Grant No. XDA2006030201), the Natural Science Foundation of Xinjiang Uygur Autonomous Region (2022D01D02) and the National Natural Science Foundation of China (Grant 31770515). Chi Zhang is supported by the Taishan Scholars Program of Shandong, China (Grant ts201712071).

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
