# Peer review of "The patterns of soil nitrogen stocks and C:N stoichiometry under impervious surfaces in China"

_Earth System Science Data, 2023_

## Author Comment (AC1)

**Response to Comments on the Manuscript (essd-2023-218):**

**The patterns of soil nitrogen stocks and C:N stoichiometry under impervious surfaces in China**

Dear Editors and Referees,

Thanks for your comments on our study "The patterns of soil nitrogen stocks and C:N stoichiometry under impervious surfaces in China" [Paper # essd-2023-218]. We have revised the manuscript accordingly and addressed your comments point by point.

Best regards,

Qian Ding, Hua Shao, Chi Zhang, Xia Fang

**RC1: 'Comment on essd-2023-218', Anonymous Referee #1, 17 Jul 2023**

**General comments:** This paper studied soil nitrogen and organic carbon stock in impervious surface areas in China. In general, this is an interesting study, which could improve our understanding of the special pattern of soil N under impervious surfaces. However, the methods used in this study and the results are not convincing at the current stage. I would suggest the authors carefully revise the manuscript based on the following comments.

**Response:** Thanks for your comments. We have revised the manuscript accordingly and addressed your comments point by point.

**Comment 1:** Abstract, line (L) 20, urbanization indeed change the permeable surface areas to impervious surface areas. Why did the urbanization not cause soil N loss?

**Response:** Thank you for point out this mistake. The original statement is incorrect. Our data showed the soil N density of impervious surfaces ($N_{ISA}$) was only about 53–69% of the national mean soil nitrogen density (2nd paragraph of section 5.1). We correct this mistake in the revised Abstract as "*The $N_{ISA}$ was also only about 53–69% of the reported national mean soil nitrogen density, indicating ISA expansion caused soil N loss.*"

We apologize for this mistake.

**Comment 2:** Figure 1, the land use type of each site can be added in the figure.

**Response:** Information of background vegetation/land-use type were extracted from the vegetation map of China (Editorial Committee of Chinese Vegetation Map, 2021) and added to the revised Figure 1 (see below). We also made additional modifications to improve the quality of Figure 1: Because the 148 sampling sites were concentrated in 41 cities, many of the site symbols overlapped and cannot be identified in the original figure. Therefore, the revised figure only shows the 41 cities with their ID numbers that

can be used for retrieving detailed information (e.g., land-use type) of the sampling sites in each city from the online dataset of this study (see Ding et al., 2023).

[Figure]

**Figure 1a: Spatial distribution of the sampled cities. The numbers in the map are the IDs of the studied cities, which can be used to retrieve detailed information of the sample sites from the online dataset of this study (Ding et al., 2023). To facilitate spatial analysis, we divided the country into six subregions – E: eastern China, S: southern China, N: northern China, NE: northeastern China, NW: northwestern China, SW: southwestern China.**

References

Editorial Committee of Chinese Vegetation Map, Chinese Academy of Sciences: 1:1 million vegetation data set in China, National Cryosphere Desert Data Center [data set], 2020.

Ding, Q., Shao, H., Zhang, C., and Fang, X.: Observations of soil nitrogen and soil organic carbon to soil nitrogen stoichiometry under the impervious surfaces areas (ISA) of China, National Cryosphere Desert Data Center, https://doi.org/10.12072/ncdc.socn.db2851.2023, 2023.

**Comment 3:** L85, what kinds of roads, elevated piers, and floor buildings? It would be great if the authors can support some pictures! I am also curious how did you take soil samples from roads, elevated piers, and floor buildings? You directly dug a soil pit

under different impervious surface areas? Is it possible for the floor buildings?

**Response:** Example photos for different type of sampling sites are added to Figure 1b. As you can see in the photos below, the samples were taken in randomly selected construction sites from ongoing engineering projects in the cities, including under the roads (b1), building floors (b2, b3) and the elevated highway piers (b4).

[Figure]

**Figure 1b: Example photos for different sampling sites.**

**Comment 4:** The unit of parameters in ALL equations should be clarified.

**Response:** We have clarified the unit of parameters in all equations in the revised manuscript:

$$N_i = \frac{NC_i \times BD_i \times 20}{100}, \qquad (1)$$
$$N_{100cm} = \sum_{i=1}^{n} N_i, \qquad (2)$$

where N represents N density (kg m$^{-2}$), $i \in [1,5]$ represents soil layer (each 20 cm in thickness), NC is N content (g kg$^{-1}$), BD is soil bulk density (g cm$^{-3}$).

$$Euclidean\ distance = \sqrt{\left(C:N_i - C:N_j\right)^2}, \qquad (3)$$

where C:N$_i$ and C;N$_j$ are the soil C:N ratios of site i and site j, respectively, when measuring the intra-city dissimilarity, or the city-averaged C:N ratios of city i and city j, respectively, when measuring the inter-city dissimilarity.

$$N_{Natural}\%_d = -0.0074d^2 + 1.7378d = (1.7378 - 0.0074d) \times d, \qquad (4)$$

where $N_{Natural}\%$ is the proportion of total N stock (in 100 cm depth) stored to depth $d$ cm in natural soil in China.

$$N_{ISA}\%_d = 1.0324d, \tag{5}$$

where $N_{ISA}\%$ $d$ (%) is the percentage of total N storage (of 100 cm depth) stored in the top $d$ (cm) depth of the soil. The unit of $N_{ISA}$ is kg m$^{-2}$.

**Comment 5:** L132-133, the citation should be formatted, and other citations in similar format should also be revised.

**Response:** We have checked the citation format and revised the related references according to the comments. Following are the revisions:

L132-133 is changed to "According to Yang et al. (2007), 46% of the N stock (in 1 m depth) of natural soil is stored in the top 0–30 cm soil, and 68% of the N stock is stored in the top 0–50 cm."

L138-139: this content has been removed in the revised manuscript.

L200-203 is changed to "To facilitate spatial analysis, we divided the country into six subregions – the northeast, north, northwest, east, south, and southwest, according to geography, climate, and socioeconomics following Ding et al. (2022) (Figure 1a)."

L227-229 is changed to "Our observed $N_{ISA}$ content (0.4 g kg$^{-1}$) in the 20–40 cm soil layer in Beijing was also comparable to the reports by Zhao et al. (2012) (0.26–0.42 g kg$^{-1}$)."

L256-260 is changed to "Similarly, Wei et al. (2014a) found that C:$N_{ISA}$ was lower than C:$N_{PSA}$ in Yixing city, China, and O'Riordan et al. (2021) found a significant positive correlation between N and C in ISA soil in Greater Manchester, UK, even though they also observed an increased total C:total N ratio in ISA soil compared to PSA soil."

L280-281 is changed to "Lu et al. (2023) found a lower C:N ratio at higher latitudes in China, suggesting a positive correlation between C:N and temperature in natural ecosystem soils."

**Figure 7: Comparing the vertical distribution pattern of N between the sealed soil (N$_{ISA}$) and the natural soil (N$_{Natural}$) in China (refer to Section 2.4 Equation 4).**

Reference:

Yang, Y., Ma, W., Mohammat, A., and Fang, J.: Storage, Patterns and Controls of Soil Nitrogen in China, Pedosphere, 17, 776-785, https://doi.org/10.1016/S1002-0160(07)60093-9, 2007.

**Comment 7:** L139-140, why did you select these parameters? Please explain and describe the detailed process of model construction.

**Response:** After considering the comment 7 and comment 8, we recognize that with the limited soil N observations and limited understanding in the N processes under impervious surfaces, it will be inappropriate to develop a N$_{ISA}$ map of China's using any spatial interpolation or modelling methods. Therefore, we decide to give up such effort and remove all the related contents in the revised manuscript (please refer to our responses to the comment 8).

Instead, we analyzed the relationships between N$_{ISA}$ and 15 potential environmental controls including climate, terrain, and social-economic factors. Following is the newly added section 2.5 in the revised manuscript that describes the selected factors and the data sources:

[revised manuscript text omitted]

**Comment 8:** Random forest is not an explainable model and might not be convincing. More methods are encouraged to be included in this manuscript, such as biogeochemistry model.

**Response:** We agree with the reviewer that using Random forest to develop a soil $N_{ISA}$ map of China is not convincing. However, we currently know little about the biogeochemical processes under the impervious surface, and cannot find appropriate biogeochemistry model to do the job. Therefore, we give up the effort in developing a national soil $N_{ISA}$ map of China, and remove the related contents in the revised manuscript. Instead, we analyzed the relationships between $N_{ISA}$ and potential

environmental controls including climate, terrain, and social-economic factors (please refer to our responses to comment 7). As shown below, the revised Section 2.5 describes the methodology; the revised Section 3.5 shows the analysis results; the revised Section 5.3 discussed the findings:

**Section 2.5: (please refer to our responses to comment 7)**

[revised manuscript text omitted]

**Comment 9:** Figure 4, the error bars of the intra-city column should also added.

**Response:** In the original manuscript, we used coefficient of variation (CV) to evaluate the variance of soil C:N among the cities (i.e., the inter-city variation in C:N). Because the mean C:N of each city was treated as one sample to calculate the CV, we are unable to estimate the uncertainty of the derived CV itself, thus unable to add an error bar.

To estimate the uncertainty in the inter-city variation in the revised manuscript, we change to use dissimilarity to quantify the variation of the C:N among the cities. The dissimilarity measured the Euclidean distance in C:N between each pair of cities. In this case, the C:N dissimilarity of each pair of cities can be treated as one sample of the variation. Their mean value is an unbiased estimate of the inter-city variation in C:N ratio and the standard deviation can be added to the chart as the error bar of the estimate. The revised figure is shown below:

[Figure]

**Figure 5: Comparing the inter–city variation and the intra–city variation of C:N ratios between the ISA and PSA. The variations were measured by the dissimilarity of (or the Euclidean distance between) paired observations. For intra–city variation, the soil C:N dissimilarity between each pair of different sampling sites within the same city were calculated and averaged; for inter-city variation, the soil C:N dissimilarity between each pair of different cities under investigation were calculated and averaged. The letters indicate the significance of the difference among the groups.**

**Comment 10:** Figure 5, the origin coordinates should also be indicated to distinguish different directions.

**Response:** Thanks for the suggestion, we added the origin coordinates to the figures.

[Figure]

**Figure 6: The variation trend of N and C:N under the two surfaces.**

**Comment 11:** In equation 4, $N_{ISA}\%$ was 2.31 when d=0, which should not be the case. Please revise the equation.

**Response:** We set the intercept to 0 and refit the linear model:

$$N_{ISA}\%_d = 1.0324d \qquad (5)$$

where $N_{ISA}\% d$ (%) is the percentage of total N storage (of 100 cm depth) stored in the top $d$ (cm) depth of the soil. The unit of $N_{ISA}$ is kg m$^{-2}$.

**Comment 12:** Figure 6, which data did you use in this figure? Why not include all of the sampling points?

**Response:** Sorry for the confusion. This figure aims to compare the vertical patterns of N between ISA soil and natural soil. Because our study area focused on urban area, we didn't collect samples in natural soil. Therefore, we relied on a pervious study (Yang et al., 2007), which compared the soil N storage to the 30 cm depth, 50 cm depth and 100

cm depth in natural soil in China, to derive the vertical pattern of N stock in natural soil (the blue line in the figure).

To prevent confusion, we added descriptions of the data sources directly in the revised figure as you can see below:

[Figure]

We also revised section 2.4 (see below) to clarify the issue:

**2.4 Investigating the vertical pattern of $N_{ISA}$**

Unlike other studies that focused on topsoil, our multiple–layer soil sampling data made it possible to study the vertical pattern of $N_{ISA}$ to a 100 cm depth. The proportions of N stored in the 0–20 cm depth, 0–40 cm depth, 0–60 cm depth, and 0–80 cm depth to the total (100 cm depth) N stock in each sample profile were calculated and plotted against the soil depth to reveal the vertical distribution pattern of $N_{ISA}$ and $N_{PSA}$. Based on these data, we could model how N storage changed with soil depth. According to Yang et al. (2007), 46% of the N stock (in 1 m depth) of natural soil is stored in the top 0–30 cm soil, and 68% of the N stock is stored in the top 0–50 cm, translating into a power function fitting model (Figure 7):

$$N_{Natural}\%_d = -0.0074d^2 + 1.7378d = (1.7378 - 0.0074d) \times d, \qquad (4)$$

where $N_{Natural}\%$ is the proportion of total N stock (in 100 cm depth) stored to depth $d$ cm in natural soil in China. The equation shows that the $N_{Natural}\%$ does not increase linearly with soil depth, its increasing rate (i.e., $1.7378 - 0.0074d$) reduces with soil depth $d$. This pattern indicates the natural soil N does not have homogeneous N density through the soil profile, it decreases with depth.

Reference:

Yang, Y., Ma, W., Mohammat, A., and Fang, J.: Storage, Patterns and Controls of Soil Nitrogen in China, Pedosphere, 17, 776-785, https://doi.org/10.1016/S1002-0160(07)60093-9, 2007.

**Comment 13:** From the data of figure 7, the predicted NISA density will be overestimated when the value is lower than ~ 0.6 kg m−2, while the opposite when the value is higher than ~ 0.6 kg m−2. The worth thing is the deviation will be much higher when the value is far away from ~ 0.6 kg m−2. Thus, I strongly suggest the authors optimize their model and the predicted value.

**Response:** This figure was removed, because we recognize that the Random Forest method is not appropriate for this study and give up the effort to develop a national soil $N_{ISA}$ map (please refer to our responses to the comment 8). Instead, we focused on analyzing the relationships between $N_{ISA}$ and the potential environmental control factors (please refer to our responses to the comment 7).

---

## Author Comment (AC2)

**Response to Comments on the Manuscript (essd-2023-218):**

**The patterns of soil nitrogen stocks and C:N stoichiometry under impervious surfaces in China**

Dear Editors and Referees,

Thanks for your comments on our study "The patterns of soil nitrogen stocks and C:N stoichiometry under impervious surfaces in China" [Paper # essd-2023-218]. We have revised the manuscript accordingly and addressed your comments point by point.

Best regards,

Qian Ding, Hua Shao, Chi Zhang, Xia Fang

**RC2: 'Comment on essd-2023-218', Anonymous Referee #2, 18 Jul 2023**

**General comments:** Ding and coauthors have conducted a national scale soil samplings and reports the patterns of soil nitrogen (N) stocks and C:N stoichiometry under impervious surfaces in China. They found that soil N density in the 0-100 cm profile under impervious surface areas was significantly lower than that under the permeable surface areas, and pointed out that the impervious surfaces could result in the convergence of soil C:N stoichiometry. Overall, this is an interesting study and provide the knowledge of biogeochemical cycles under impervious surfaces. However, I have several concerns on the present manuscript.

**Response:** Thanks for your comments. We have revised the manuscript accordingly and addressed your comments point by point.

**Comment 1: Introduction:** All previous studies have already demonstrated that impervious surface areas had lower soil N density than permeable surface areas, what is the novelty of comparing soil N density under impervious surface areas with that under permeable surface areas (In the present study, the authors also found that soil N density under impervious surface areas was significantly lower than that under the permeable surface areas)? Moreover, even though there is a lack of information of vertical variations in soil N densities under impervious surface areas, the authors should introduce the necessity of studying vertical distributions of soil N and should be better to propose the hypothesis (is it different from that in natural soils or the soils under permeable surface areas)? I would suggest the authors further improve the novelty and significance of their study. In addition, the data in the sentence "ISA covers …… during 2000-2030" is pretty old, please use the updated information. I am also confused with the expression in the sentence "We chose to use …… from construction materials", to the best of my knowledge, soil C:N stoichiometry represents the SOC:total N ratio rather than the total C:total N ratio, why did the authors state an information different from the common sense?

**Response:**

(1) The fourth paragraph of the Introduction section has been completely revised to highlight the novelty and significance of this study. We emphasized that previous studies focused on individual cities, thus were unable to gain a big picture of the large-scale distribution pattern of the soil N in impervious surface area ($N_{ISA}$). This limitation also makes it impossible to evaluate the correlation between $N_{ISA}$ and the potential environmental drivers such as climate factors, geographic factors, and socio-economic factors, etc.. Finally, our literature review can only find 7 local scale case studies so far, which is far from adequate to estimate the $N_{ISA}$ pool size at large scale. Following is the revised paragraph:

Considering the high heterogeneity of urban soils, the available observations from 7 cities around the world are far from enough to provide useful information about the storage and characteristic distribution of $N_{ISA}$ at large scale. Previous studies focused on individual cities, but regional scale surveys are required to investigate the influences of climatic, ecological, geographic, and socioeconomic factors on $N_{ISA}$ distribution. Such information is not only necessary to evaluate global $N_{ISA}$ pool size, but also helpful in revealing the environmental-control mechanisms over the soil biogeochemical processes in ISA (Ding et al., 2022). For example, the urban ecosystem convergence theory suggests that cities from different regions tend to have similar soil properties (e.g., SOC density) as a result of intensive human disturbances, even if their native soil properties differ significantly (Pouyat et al., 2003). Regional soil surveys from multiple cities are required to evaluate this theory with soil nutrient data. In addition, more observational data are required to evaluate whether ISA soil has extremely high C:N ratio, which might indicate decoupling of soil C and N processes (Raciti et al., 2012; O'riordan et al., 2021).

(2) We added the following paragraph to emphasize the importance of study the vertical pattern of soil N under the ISA:

Investigations on the vertical distribution pattern of soil N are also important, because the nutrient distribution patterns through soil profiles are influenced by both natural and human factors. In natural ecosystems, vertical nutrient distributions are dominated by plant cycling relative to leaching, weathering dissolution, and atmospheric deposition, leading to nutrient concentrating in topsoil (Jobbágy and Jackson, 2001). Previous studies in urban areas, however, showed that the removal of plants and topsoil in the ISA may alter the vertical pattern of SOC, resulting in a more homogeneous SOC distribution through the soil profile (Yan et al., 2015; Ding et al., 2022). Based on the observed SOC pattern, previous studies suggested that the changes in soil biogeochemistry in ISA was mainly caused by plant and topsoil removals and initial disturbance as opposed to postconstruction processes (Jobbágy and Jackson, 2001). Investigations on the vertical distribution patterns of $N_{ISA}$ can help us to evaluate this mechanism. However, most previous studies only sampled the topsoil or upper soil layers (Table 1) and thus could not obtain a complete picture of the vertical distribution pattern of the $N_{ISA}$.

(3) We updated the information of global ISA expansion in the revised manuscript:

The global ISA area in 2018 was 1.5 times larger than in 1990, at approximately $7.97 \times 10^5$ km$^2$ (Gong et al., 2020)

(4) We agree that it is widely accepted that soil C:N stoichiometry represents the SOC:total N ratio rather than the total C:total N ratio. However, we noticed that some previous research (Hu et al., 2018; Pereira et al., 2021; O'riordan et al., 2021) used the ratio between total C and total N to investigate the C:N stoichiometry in ISA soil.

To prevent confusion, we changed all SOC:N to C:N in the revised manuscript, and discussed the issue in the revised section 2.3 (paragraph 2, see below):

[revised manuscript text omitted]

**Comment 2: Materials and methods:** A big concern is the calculation of soil N density, why the authors did not consider the rock fragments in calculating the soil N density? For calculating the soil C and N density, using the fine earth bulk density and soil N concentration can provide more accurate N density estimation. In addition, there are some mistakes for the citation formats, for example, Yang et al. (2007) (Yang et al., 2007), Zhang et al. (2021) (Zhang et al., 2021), the authors should treat their manuscript more carefully throughout the whole manuscript.

**Response:**

(1) Sorry for fail to described our soil sampling method in detail in the manuscript. Yes, we tried to exclude large amount of rock fragments in soil sampling treatment. As you can see below, we added detailed description of our sampling treatment methodology in the revised section 2.1:

Our study across China found that most of the Ekranic (sealed) Technosol profiles have a clear boundary between the building material layer and the soil. Where the boundary is unclear, we treated the topsoil with a high amount of hard building materials, where artifacts >0.15 mm accounted for over half of the soil volume, as the building material layer. We only took samples in the soil below the building material layer. Samples with notable additions of anthropogenic artifacts, e.g., coal fly ash, mixed in the soil were discarded. Following the protocol of China's National Soil Surveys, the visible non-soil artifacts in the remaining soil samples, such as fragmentations of bricks, glasses, stones, roots, etc., were picked out and discarded (Shi et al., 2004).

(2) We have checked the citation formats and revised the related references according to the comments. Following are the corrected citations.

L132-133 is changed to "According to Yang et al. (2007), 46% of the N stock (in 1 m depth) of natural soil is stored in the top 0–30 cm soil, and 68% of the N stock is stored in the top 0–50 cm."

L138-139: this content has been removed from the manuscript.

L200-203 is changed to "To facilitate spatial analysis, we divided the country into six subregions – the northeast, north, northwest, east, south, and southwest, according to geography, climate, and socioeconomics following Ding et al. (2022) (Figure 1a)."

L227-229 is changed to "Our observed $N_{ISA}$ content (0.4 g kg$^{-1}$) in the 20–40 cm soil layer in Beijing was also comparable to the reports by Zhao et al. (2012) (0.26–0.42 g kg$^{-1}$)."

L256-260 is changed to "Similarly, Wei et al. (2014a) found that C:$N_{ISA}$ was lower than C:$N_{PSA}$ in Yixing city, China, and O'Riordan et al. (2021) found a significant positive correlation between N and C in ISA soil in Greater Manchester, UK, even though they also observed an increased total C:total N ratio in ISA soil compared to PSA soil."

L280-281 is changed to "Lu et al. (2023) found a lower C:N ratio at higher latitudes in China, suggesting a positive correlation between C:N and temperature in natural ecosystem soils."


Figure 5 shows the ISA soil samples had lower inter-city C:N dissimilarity (1.86±1.40 vs. 2.43±2.15) than PSA, but similar intra-city C:N dissimilarity to PSA. This pattern indicates that although the ISA soil and the PSA soil had similar variations in C:N stoichiometry at the local scale (within a city), the C:N variations at national scale (among the cities) were reduced for the ISA soil, possibly due to the intensive human disturbances on ISA soil as predicted by the urban ecosystem convergence theory (Pouyat et al., 2003).

(3) Finally, we revised the caption of the figure to clarify the issue:

Figure 5: Comparing the inter–city variation and the intra–city variation of C:N ratios between the ISA and PSA. The variations were measured using the dissimilarity of (or the Euclidean distance between) paired observations. For intra–city variation, the soil C:N dissimilarity between each pair of different sampling sites within the same city were calculated and averaged; for inter-city variation, the soil C:N dissimilarity between each pair of different cities under investigation were calculated and averaged. The letters indicate the significance of the difference among the groups.

---

## Author Response (AR1)

**The patterns of soil nitrogen stocks and C:N stoichiometry under impervious surfaces in China**

Dear Editors and Referees,

Thanks for your comments on our study "The patterns of soil nitrogen stocks and C:N stoichiometry under impervious surfaces in China" [Paper # essd-2023-218]. We have revised the manuscript accordingly and addressed your comments point by point.

Best regards,

Qian Ding, Hua Shao, Chi Zhang, Xia Fang

**RC1: 'Comment on essd-2023-218', Anonymous Referee #1, 17 Jul 2023**

**General comments:** This paper studied soil nitrogen and organic carbon stock in impervious surface areas in China. In general, this is an interesting study, which could improve our understanding of the special pattern of soil N under impervious surfaces. However, the methods used in this study and the results are not convincing at the current stage. I would suggest the authors carefully revise the manuscript based on the following comments.

**Response:** Thanks for your comments. We have revised the manuscript accordingly and addressed your comments point by point.

**Comment 1:** Abstract, line (L) 20, urbanization indeed change the permeable surface areas to impervious surface areas. Why did the urbanization not cause soil N loss?

**Response:** Thank you for point out this mistake. The original statement is incorrect. Our data showed the soil N density of impervious surfaces ($N_{ISA}$) was only about 53–69% of the national mean soil nitrogen density (2nd paragraph of section 5.1). We correct this mistake in the revised Abstract as "*The $N_{ISA}$ was also only about 53–69% of the reported national mean soil nitrogen density, indicating ISA expansion caused soil N loss.*" (Line 19–Line 20 in the revised manuscript)

We apologize for this mistake.

**Comment 2:** Figure 1, the land use type of each site can be added in the figure.

**Response:** Information of background vegetation/land-use type were extracted from the vegetation map of China (Editorial Committee of Chinese Vegetation Map, 2021) and added to the revised Figure 1 (see below) (Line 100–Line 108 in the revised manuscript). We also made additional modifications to improve the quality of Figure 1: Because the 148 sampling sites were concentrated in 41 cities, many of the site symbols overlapped and cannot be identified in the original figure. Therefore, the revised figure

only shows the 41 cities with their ID numbers that can be used for retrieving detailed information (e.g., land-use type) of the sampling sites in each city from the online dataset of this study (see Ding et al., 2023).

[Figure]

**Figure 1a: Spatial distribution of the sampled cities. The numbers in the map are the IDs of the studied cities, which can be used to retrieve detailed information of the sample sites from the online dataset of this study (Ding et al., 2023). The background land-use type shows the regional dominant land-use/land-cover type where the cities locates. To facilitate spatial analysis, we divided the country into six subregions – E: eastern China, S: southern China, N: northern China, NE: northeastern China, NW: northwestern China, SW: southwestern China.**

References

Editorial Committee of Chinese Vegetation Map, Chinese Academy of Sciences: 1:1 million vegetation data set in China, National Cryosphere Desert Data Center [data set], 2020.

Ding, Q., Shao, H., Zhang, C., and Fang, X.: Observations of soil nitrogen and soil organic carbon to soil nitrogen stoichiometry under the impervious surfaces areas (ISA) of China, National Cryosphere Desert Data Center, https://doi.org/10.12072/ncdc.socn.db2851.2023, 2023.

**Comment 3:** L85, what kinds of roads, elevated piers, and floor buildings? It would be

great if the authors can support some pictures! I am also curious how did you take soil samples from roads, elevated piers, and floor buildings? You directly dug a soil pit under different impervious surface areas? Is it possible for the floor buildings?

**Response:** Example photos for different type of sampling sites are added to Figure 1b (Line 100–Line 108 in the revised manuscript). As you can see in the photos below, the samples were taken in randomly selected construction sites from ongoing engineering projects in the cities, including under the roads (b1), building floors (b2, b3) and the elevated highway piers (b4).

[Figure]

**Figure 1b: Example photos for different sampling sites.**

**Comment 4:** The unit of parameters in ALL equations should be clarified.

**Response:** We have clarified the unit of parameters in all equations in the revised manuscript:

$$N_i = \frac{NC_i \times BD_i \times 20}{100}, \tag{1}$$
$$N_{100cm} = \sum_{i=1}^{n} N_i, \tag{2}$$

where N represents N density (kg m$^{-2}$), $i \in [1,5]$ represents soil layer (each 20 cm in thickness), NC is N content (g kg$^{-1}$), BD is soil bulk density (g cm$^{-3}$). (Line 134–Line 137 in the revised manuscript)

$$\text{Euclidean distance} = \sqrt{\left(C{:}N_i - C{:}N_j\right)^2}, \tag{3}$$

where C:N$_i$ and C:N$_j$ are the soil C:N ratios of site i and site j, respectively, when measuring the intra-city dissimilarity, or the city-averaged C:N ratios of city i and city j, respectively, when measuring the inter-city dissimilarity. (Line 161–Line 164 in the revised manuscript)

$$N_{Natural}\%_d = -0.0074d^2 + 1.7378d = (1.7378 - 0.0074d) \times d, \qquad (4)$$

where $N_{Natural}\%$ is the proportion of total N stock (in 100 cm depth) stored to depth $d$ cm in natural soil in China. (Line 173–Line 175 in the revised manuscript)

$$N_{ISA}\%_d = 1.0324d, \qquad (5)$$

where N$_{ISA}\%$ $d$ (%) is the percentage of total N storage (of 100 cm depth) stored in the top $d$ (cm) depth of the soil. The unit of N$_{ISA}$ is kg m$^{-2}$. (Line 264–Line 266 in the revised manuscript)

**Comment 5:** L132-133, the citation should be formatted, and other citations in similar format should also be revised.

**Response:** We have checked the citation format and revised the related references according to the comments. Following are the revisions:

L132-133 is changed to "According to Yang et al. (2007), 46% of the N stock (in 1 m depth) of natural soil is stored in the top 0–30 cm soil, and 68% of the N stock is stored in the top 0–50 cm." (Line 170–Line 172 in the revised manuscript)

L138-139: this content has been removed in the revised manuscript.

L200-203 is changed to "To facilitate spatial analysis, we divided the country into six subregions – the northeast, north, northwest, east, south, and southwest, according to geography, climate, and socioeconomics following Ding et al. (2022) (Figure 1a)." (Line 225–Line 227 in the revised manuscript)

L227-229 is changed to "Our observed N$_{ISA}$ content (0.4 g kg$^{-1}$) in the 20–40 cm soil layer in Beijing was also comparable to the reports by Zhao et al. (2012) (0.26–0.42 g kg$^{-1}$)." (Line 296–Line 298 in the revised manuscript)

L256-260 is changed to "Similarly, Wei et al. (2014a) found that C:N$_{ISA}$ was lower than C:N$_{PSA}$ in Yixing city, China, and O'Riordan et al. (2021) found a significant positive correlation between N and C in ISA soil in Greater Manchester, UK, even though they also observed an increased total C:total N ratio in ISA soil compared to PSA soil." (Line 326–Line 329 in the revised manuscript)

L280-281 is changed to "Lu et al. (2023) found a lower C:N ratio at higher latitudes in China, suggesting a positive correlation between C:N and temperature in natural ecosystem soils." (Line 361–Line 362 in the revised manuscript)

**Figure 7: Comparing the vertical distribution pattern of N between the sealed soil ($N_{ISA}$) and the natural soil ($N_{Natural}$) in China (refer to Section 2.4 Equation 4).**

Reference:

Yang, Y., Ma, W., Mohammat, A., and Fang, J.: Storage, Patterns and Controls of Soil Nitrogen in China, Pedosphere, 17, 776-785, https://doi.org/10.1016/S1002-0160(07)60093-9, 2007.

**Comment 7:** L139-140, why did you select these parameters? Please explain and describe the detailed process of model construction.

**Response:** After considering the comment 7 and comment 8, we recognize that with the limited soil N observations and our limited understanding in the N processes under impervious surfaces, it will be inappropriate to develop a $N_{ISA}$ map of China using any spatial interpolation or modelling methods. Therefore, we decide to give up such effort and remove all the related contents in the revised manuscript (please refer to our responses to the comment 8).

Instead, we analyzed the relationships between $N_{ISA}$ and 15 potential environmental controls including climate, terrain, and social-economic factors. Following is the newly added section 2.5 in the revised manuscript that describes the selected factors and the data sources (Line 178–Line 210 in the revised manuscript):

[revised manuscript text omitted]

**Comment 8:** Random forest is not an explainable model and might not be convincing. More methods are encouraged to be included in this manuscript, such as biogeochemistry model.

**Response:** We agree with the reviewer that using Random forest to develop a soil $N_{ISA}$ map of China is not convincing. However, we currently know little about the

biogeochemical processes under the impervious surface, and cannot find an appropriate biogeochemistry model to do the job. Therefore, we give up the effort in developing a national soil $N_{ISA}$ map of China, and remove the related contents in the revised manuscript. Instead, we analyzed the relationships between $N_{ISA}$ and potential environmental controls including climate, terrain, and social-economic factors (please refer to our responses to the comment 7). As shown below, the revised Section 2.5 (Line 178–Line 210 in the revised manuscript) describes the methodology; the revised Section 3.5 (Line 269–Line 283 in the revised manuscript) shows the analysis results; the revised Section 5.3 (Line 337–Line 372 in the revised manuscript) discusses the findings:

**Section 2.5: (please refer to our above responses to comment 7)**

[revised manuscript text omitted]

**Comment 9:** Figure 4, the error bars of the intra-city column should also added.

**Response:** In the original manuscript, we used coefficient of variation (CV) to evaluate the variance of soil C:N among the cities (i.e., the inter-city variation in C:N). Because the mean C:N of each city was treated as one sample to calculate the CV, we are unable to estimate the uncertainty of the derived CV itself, thus unable to add an error bar. However, we agree with the reviewer that it's better to also estimate the uncertainty in the inter-city variation.

To estimate the uncertainty in the inter-city variation in the revised manuscript, we change to use dissimilarity to quantify the variation of the C:N among the cities. The dissimilarity measured the Euclidean distance in C:N between each pair of cities. In this case, the C:N dissimilarity of each pair of cities can be treated as one sample of the variation. Their mean value is an unbiased estimate of the inter-city variation in C:N ratio and the standard deviation can be added to the chart as the error bar of the estimate. A description of the method is provided in the revised methodology section (Line 154-line164). The revised figure is shown below (Line 243–Line 248 in the revised manuscript):

[Figure]

**Figure 5: Comparing the inter–city variation and the intra–city variation of C:N ratios between the ISA and PSA. The variations were measured by the dissimilarity of (or the Euclidean distance between) paired observations. For intra–city variation, the soil C:N dissimilarity between each pair of different sampling sites within the same city were calculated and averaged; for inter-city variation, the soil C:N dissimilarity between each pair of different cities under investigation were calculated and averaged. The letters indicate the significance of the difference among the groups.**

**Comment 10:** Figure 5, the origin coordinates should also be indicated to distinguish different directions.

**Response:** Thanks for the suggestion, we added the origin coordinates to the figures (Line 258 in the revised manuscript).

[Figure]

**Figure 6: The variation trend of N and C:N under the two surfaces.**

**Comment 11:** In equation 4, $N_{ISA}\%$ was 2.31 when d=0, which should not be the case. Please revise the equation.

**Response:** We set the intercept to 0 and refit the linear model (Line 264–Line 266 in the revised manuscript):

$$N_{ISA}\%_d = 1.0324d, \tag{5}$$

where $N_{ISA}\%$ $d$ (%) is the percentage of total N storage (of 100 cm depth) stored in the top $d$ (cm) depth of the soil. The unit of $N_{ISA}$ is kg m$^{-2}$.

**Comment 12:** Figure 6, which data did you use in this figure? Why not include all of the sampling points?

**Response:** Sorry for the confusion. This figure aims to compare the vertical patterns of N between ISA soil and natural soil. Because our study area focused on urban area, we didn't collect samples in natural soils. Therefore, we relied on a pervious study (Yang

et al., 2007), which compared the soil N storage to the 30 cm depth, 50 cm depth and 100 cm depth in natural soil in China, to derive the vertical pattern of N stock in natural soil (the blue line in the figure).

To prevent confusion, we added descriptions of the data sources directly in the revised figure as you can see below (Line 267–Line 268 in the revised manuscript):

[Figure]

We also revised section 2.4 (see below) (Line 165–Line 177 in the revised manuscript) to clarify the issue:

**2.4 Investigating the vertical pattern of $N_{ISA}$**

Unlike other studies that focused on topsoil, our multiple–layer soil sampling data made it possible to study the vertical pattern of $N_{ISA}$ to a 100 cm depth. The proportions of N stored in the 0–20 cm depth, 0–40 cm depth, 0–60 cm depth, and 0–80 cm depth to the total (100 cm depth) N stock in each sample profile were calculated and plotted against the soil depth to reveal the vertical distribution pattern of $N_{ISA}$ and $N_{PSA}$. Based on these data, we could model how N storage changed with soil depth. According to Yang et al. (2007), 46% of the N stock (in 1 m depth) of natural soil is stored in the top 0–30 cm soil, and 68% of the N stock is stored in the top 0–50 cm, translating into a power function fitting model (Figure 7):

$$N_{Natural}\%_d = -0.0074d^2 + 1.7378d = (1.7378 - 0.0074d) \times d, \quad (4)$$

where $N_{Natural}\%$ is the proportion of total N stock (in 100 cm depth) stored to depth $d$ cm in natural soil in China. The equation shows that the $N_{Natural}\%$ does not increase linearly with soil depth, its increasing rate (i.e., $1.7378 - 0.0074d$) reduces with soil depth $d$. This pattern indicates the natural soil N does not have homogeneous N density through the soil profile, it decreases with depth.


Figure 5 shows the ISA soil samples had lower inter-city C:N dissimilarity (1.86±1.40 vs. 2.43±2.15)

than PSA, but similar intra-city C:N dissimilarity to PSA. This pattern indicates that although the ISA soil and the PSA soil had similar variations in C:N stoichiometry at the local scale (within a city), the C:N variations at national scale (among the cities) were reduced for the ISA soil, possibly due to the intensive human disturbances on ISA soil as predicted by the urban ecosystem convergence theory (Pouyat et al., 2003).

(3) Finally, we revised the caption of the figure to clarify the issue (Line 243–Line 248 in the revised manuscript):

Figure 5: Comparing the inter–city variation and the intra–city variation of C:N ratios between the ISA and PSA. The variations were measured using the dissimilarity of (or the Euclidean distance between) paired observations. For intra–city variation, the soil C:N dissimilarity between each pair of different sampling sites within the same city were calculated and averaged; for inter-city variation, the soil C:N dissimilarity between each pair of different cities under investigation were calculated and averaged. The letters indicate the significance of the difference among the groups.

---

## Author Response (AR2)

**Response to Comments on the Manuscript (essd-2023-218):**

**The patterns of soil nitrogen stocks and C:N stoichiometry under impervious surfaces in China**

Dear Editors and Referees,

Thanks for your comments on our study "The patterns of soil nitrogen stocks and C:N stoichiometry under impervious surfaces in China" [Paper # essd-2023-218]. We have revised the manuscript accordingly and addressed your comments point by point.

Best regards,

Qian Ding, Hua Shao, Chi Zhang, Xia Fang

**'Report #1 on essd-2023-218', Anonymous Referee #1**

**General comments:** The authors have addressed most of my concerns. I only have few comments need to be considered before it can be published.

**Response:** Thanks for your comments. We have revised the manuscript accordingly and addressed your comments point by point.

**Comment 1:** Table 1, delete the vertical lines; for the data of Beijing, delete repeated data of this study; please also include the information of ISA type.

**Response:** As recommended by the reviewer, we have deleted the vertical lines and the repeated data of this study of Beijing in the revised Table 1 (see below). However, it is difficult to add information of ISA type for each city, because Table 1 shows the city-averaged soil N density/content of multiple samples from various ISA types (to prevent confusion, we revised the related descriptions from "N density/content" to "Mean observed N density/content in the city" in the revised Table 1). It is not very meaningful to list all the sampled ISA types in each city. Instead, we added the information of background land-use type where the cities locate. (Line 99–Line 100 in the latest revised manuscript). As mentioned in Line 108, detailed descriptions of the ISA type for each sample stie can be found in our online dataset (Ding et al., 2023).

**Table 1: Compilation of soil $N_{ISA}$ studies**

| City, country | Previous studies | | | | This study | | | Background land-use type |
|---|---|---|---|---|---|---|---|---|
| | Mean observed N density in the city (kg m$^{-2}$) | Mean observed N content in the city (g kg$^{-1}$) | Depth (cm) | References | Mean observed N density in the city (kg m$^{-2}$) | Mean observed N content in the city (g kg$^{-1}$) | Depth (cm) | |
| Beijing, China | NA | 0.61 | 0–10 | (Zhao et al., 2012) | 0.08±0.02 | 0.34±0.06 | 0–20 | Cropland and deciduous orchards |
| | NA | 0.54 | 10–20 | | | | | |
| | NA | 0.42 | 20–30 | | 0.09±0.02 | 0.4±0.11 | 20–40 | |
| | NA | 0.26 | 30–40 | | 0.09±0.02 | 0.4±0.11 | 20–40 | |
| | NA | 0.37 | 0–15 | (Hu et al., 2018) | 0.08±0.02 | 0.34±0.06 | 0–20 | |
| Nanjing, China | NA | 0.49 | 0–20 | (Wei et al., 2014b) | 0.38±0.05 | 0.13±0.15 | 0–20 | |
| Yixing, China | 0.25 | NA | 0–20 | (Wei et al., 2014a) | 0.15±0.01 | NA | 0–20 | NA |

| | | | | | | | | |
|---|---|---|---|---|---|---|---|---|
| New York, USA | 0.014 | NA | 0–15 | (Raciti et al., 2012) | 0.10±0.06 | NA | 0–15 | NA |
| Lancaster, UK | NA | 2.08 | 0–10 | (Pereira et al., 2021) | 0.07±0.04 | NA | 0–10 | NA |
| Greater Manchester, UK | 0.081 | NA | 0–10 | (O'riordan et al., 2021) | 0.07±0.04 | NA | 0–10 | NA |
| Toruń, Poland | 0.027 | 0.17 | 15–25 or 10–20 | (Piotrowska-Długosz and Charzyński, 2015) | 0.12±0.08 | NA | 0–20 | NA |

*±1SD

References

Zhao, D., Li, F., Wang, R., Yang, Q., and Ni, H.: Effect of soil sealing on the microbial biomass, N transformation and related enzyme activities at various depths of soils in urban area of Beijing, China, J. Soils Sediments, 12, 519-530, https://doi.org/10.1007/s11368-012-0472-6, 2012.

Hu, Y., Dou, X., Li, J., and Li, F.: Impervious Surfaces Alter Soil Bacterial Communities in Urban Areas: A Case Study in Beijing, China, Frontiers in Microbiology, 9, https://doi.org/10.3389/fmicb.2018.00226, 2018.

Wei, Z., Wu, S., Zhou, S., Li, J., and Zhao, Q.: Soil Organic Carbon Transformation and Related Properties in Urban Soil Under Impervious Surfaces, Pedosphere, 24, 56-64, https://doi.org/10.1016/s1002-0160(13)60080-6, 2014b.

Wei, Z., Wu, S., Yan, X., and Zhou, S.: Density and Stability of Soil Organic Carbon beneath Impervious Surfaces in Urban Areas, Plos One, 9, https://doi.org/10.1371/journal.pone.0109380, 2014a.

Raciti, S. M., Hutyra, L. R., and Finzi, A. C.: Depleted soil carbon and nitrogen pools beneath impervious surfaces, Environmental Pollution, 164, 248-251, https://doi.org/10.1016/j.envpol.2012.01.046, 2012.

Pereira, M. C., O'Riordan, R., and Stevens, C.: Urban soil microbial community and microbial-related carbon storage are severely limited by sealing, J. Soils Sediments, 21, 1455-1465, https://doi.org/10.1007/s11368-021-02881-7, 2021.

O'Riordan, R., Davies, J., Stevens, C., and Quinton, J. N.: The effects of sealing on urban soil carbon and nutrients, SOIL, 7, 661-675, https://doi.org/10.5194/soil-7-661-2021, 2021.

Piotrowska-Długosz, A. and Charzyński, P.: The impact of the soil sealing degree on microbial biomass, enzymatic activity, and physicochemical properties in the Ekranic Technosols of Toruń (Poland), J. Soils Sediments, 15, 47-59, https://doi.org/10.1007/s11368-014-0963-8, 2015.

Ding, Q., Shao, H., Zhang, C., and Fang, X.: Observations of soil nitrogen and soil organic carbon to soil nitrogen stoichiometry under the impervious surfaces areas (ISA) of China, National Cryosphere Desert Data Center, https://doi.org/10.12072/ncdc.socn.db2851.2023, 2023.

**Comment 2:** L146, what do you mean black C?

**Response:** We explained the term black C in Line 153–Line 155: Black C is "soot or

carbonaceous products formed during the incomplete combustion of biomass and fossil fuels" (He and Zhang, 2009; Zhu et al., 2019). It can be a significant component of some urban soils (e.g., He and Zhang, 2009).

References

He, Y. and Zhang, G. L.: Historical record of black carbon in urban soils and its environmental implications, Environmental Pollution, 157, 2684-2688, https://doi.org/10.1016/j.envpol.2009.05.019, 2009.

Zhu, M., Li, M., Wei, S., Song, J., Hu, J., Jia, W., and Peng, P. a.: Evaluation of a dichromate oxidation method for the isolation and quantification of black carbon in ancient geological samples, Organic Geochemistry, 133, 20-31, https://doi.org/10.1016/j.orggeochem.2019.03.009, 2019.

**Comment 3:** L172, change Figure 7 to Figure 1.

**Response:** The reference to Figure 7 in Line 172 is removed to prevent confusion.

**Comment 4:** L181-190, these sentences can be moved to Introduction.

**Response:** We have moved these sentences to Introduction (Line 60–Line 68 in the latest revised manuscript).

**Comment 5:** L197-201, Figure 2?

**Response:** Sorry for the mistakes. The errors were corrected in Line 197–Line 201:

Gridded datasets of environmental factors, including mean annual temperature (Figure 2a), annual precipitation (Figure 2b), and elevation (Figure 2d) at 1 km resolution, were obtained from the Data Center for Resource and Environmental Sciences, Chinese Academy of Sciences (http://www.resdc.cn/). The national NPP (1985–2015) estimates at 1 km resolution was obtained from the Digital Journal of Global Change Data Repository (https://www.geodoi.ac.cn/) (Figure 2c).

**Comment 6:** L204, please confirm the data are normal distributed and linear correlated, then the Pearson's correlation can be applied. Otherwise, Spearman's correlation should be used.

**Response:** Thank you for the reminder. We have examined the data distribution. If the variables were normal distributed and linear correlated, then the Pearson's correlation were applied. Otherwise, Spearman's correlation were used in the revised manuscript. Our analysis showed that only the Latitude, temperature, NPP, and C:N$_{ISA}$ were non-normally distributed, so we used Spearman's correlation to analysis their correlationships with N$_{ISA}$. (Line 205–Line 206 and Line 278–Line 280 in the latest revised manuscript)

**Comment 7:** Figures 3 and 4 can be merged together.

**Response:** We have merged Figures 3 and Figure 4 (see below). (Line 222–Line 227 in the latest revised manuscript)

[Figure]

**Figure 3: The N density, N storage and C:N ratio of the ISA in China. The soil N density and**

**C:N in the ISA and the reference PSA are compared in (a) and (b), respectively. The square box shows median and quad values, the inner small rectangle is the mean value. The regional mean $N_{ISA}$ density and $N_{PSA}$ density in different subregions are compared in (c). The regional $N_{ISA}$ storage of different subregions are compared in (d). The letters indicate the significance of the difference among the subregions. \* and \*\* indicate significant differences between ISA and PSA, $p<0.05$ and $p<0.01$, respectively.**

**Comment 8:** Figure 6, the authors should clarify the meaning of the figure, and how to achieve the simulation lines.

**Response:** We have provided detailed explanation of the trend analysis charts in the revised caption of Figure 6 (Line 256–Line 266 in the latest revised manuscript)

[Figure]

**Figure 5: Trend analysis on the variations of the city-level mean (a) $N_{ISA}$ density, (b) $N_{PSA}$ density, (c) C:$N_{ISA}$ and (d) C:$N_{PSA}$, in the east–west direction (the green trend line) and the north-south direction (the blue trend line) across China. The locations of sampled cities are plotted on the x, y plane. The x-axis indicates the east-west direction; the y-axis indicates the north-south direction. Above each sampled city, its city-averaged value of observed soil N property (i.e., $N_{ISA}$ density or $N_{PSA}$ density or C:N ratio) is given by the height of a stick in the z-dimension. The values are projected onto the x, z plane (i.e., the left vertical plane) and the**

**y, z plane (i.e., the right vertical plane) as scatterplots. This can be thought of as sideways views through the three-dimensional data. Second-order polynomials are fit through the scatterplots on the projected planes. The green line in the x, z plane shows the trend of value variation in the east-west direction, while the blue line in the y, z plane shows the trend of value variation in the north-south direction.**

**Comment 9:** L265, the sentence need to be revised.

**Response:** The sentence was revised to:

where $N_{ISA}\%_d$ is the percentage of the N stock (to 100 cm depth in total) located in the top $d$ (cm) depth of the soil. (Line 273–Line 274 in the latest revised manuscript)

**Comment 10:** Table 2, modify the format; person km-2; person 10-4 yuan.

**Response:** Thanks for the suggestion. We have modified the format. We also made additional modifications to improve the art quality of the Table. (Line 290–Line 292 in the latest revised manuscript)

**Table 2: Correlations between $N_{ISA}$, C:$N_{ISA}$ and potential environmental drivers**

| Factors | N density (kg m$^{-2}$) | | C:$N_{ISA}$ | |
|---|---|---|---|---|
| | Correlation Coefficient | Sig. (2 tailed) | Correlation Coefficient | Sig. (2 tailed) |
| Longitude | 0.196 | 0.22 | –0.186 | 0.24 |
| Latitude | 0.275 | 0.08 | 0.513$^{**}$ | 0.00 |
| DEM (m) | 0.141 | 0.38 | 0.477$^{**}$ | 0.00 |
| Annual precipitation (mm) | –0.268 | 0.09 | –0.620$^{**}$ | 0.00 |
| Mean Temperature (°C) | –0.486$^{**}$ | 0.00 | –0.561$^{**}$ | 0.00 |
| NPP (g m$^{-2}$) | –0.106 | 0.51 | –0.516$^{**}$ | 0.00 |
| ISA coverage in built-up area (%) | –0.126 | 0.43 | –0.171 | 0.29 |
| Built-up area (km$^2$) | –0.072 | 0.65 | 0.062 | 0.70 |
| Greenspace coverage in built-up area (%) | –0.229 | 0.15 | –0.063 | 0.69 |
| Population density (person km$^{-2}$) | –0.032 | 0.84 | –0.072 | 0.66 |
| Per capita GDP (person $10^{-4}$ yuan) | –0.012 | 0.94 | –0.145 | 0.37 |
| City GDP (billion yuan) | –0.015 | 0.93 | –0.200 | 0.21 |

| | | | | |
|---|---|---|---|---|
| Per capita greenspace ($m^2$) | 0.098 | 0.54 | 0.044 | 0.79 |
| The fraction of the newly expanded ISA since 2002 (%) | $-0.340^*$ | 0.03 | $-0.197$ | 0.22 |
| $N_{PSA}$ density (kg $m^{-2}$) | $0.715^{**}$ | 0.00 | NA | NA |
| C:$N_{PSA}$ | NA | NA | $0.515^{**}$ | 0.00 |
| BD | $-0.104$ | 0.52 | NA | NA |

$^*$p < 0.05;
$^{**}$p < 0.01.